# Genome-wide identification and characterization of FORMIN gene family in potato (*Solanum tuberosum* L.) and their expression profiles in response to drought stress condition

**Mst. Sumaiya Khatun**[1], **Md Shohel Ul Islam**[1], **Pollob Shing**[1], **Fatema Tuz Zohra**[2], **Shuraya Beente Rashid**[1], **Shaikh Mizanur Rahman**[1], **Md. Abdur Rauf Sarkar**[1]*

1 Laboratory of Functional Genomics and Proteomics, Department of Genetic Engineering and Biotechnology, Faculty of Biological Science and Technology, Jashore University of Science and Technology, Jashore, Bangladesh, 2 Department of Genetic Engineering and Biotechnology, Faculty of Biological Sciences, University of Rajshahi, Rajshahi, Bangladesh

☯ These authors contributed equally to this work.
* rauf.gebt@yahoo.com

**Data Availability Statement:** All relevant data are within the manuscript and its Supporting Information files.

## Abstract

Formin proteins, characterized by the FH2 domain, are critical in regulating actin-driven cellular processes and cytoskeletal dynamics during abiotic stress. However, no genome-wide analysis of the formin gene family has yet to be conducted in the economically significant plant potato (*Solanum tuberosum* L.). In this study, 26 formin genes were identified and characterized in the potato genome (named as *StFH*), each containing the typical FH2 domain and distributed across the ten chromosomes. The *StFH* was categorized into seven subgroups (A-G) and the gene structure and motif analysis demonstrated higher structural similarities within the subgroups. Besides, the *StFH* exhibited ancestry and functional similarities with *Arabidopsis*. The Ka/Ks ratio indicated that *StFH* gene pairs were evolving through purifying selection, with five gene pairs exhibiting segmental duplications and two pairs exhibiting tandem duplications. Subcellular localization analysis suggested that most of the *StFH* genes were located in the chloroplast and plasma membrane. Moreover, 54 *cis*-acting regulatory elements (CAREs) were identified in the promoter regions, some of which were associated with stress responses. According to gene ontology analysis, the majority of the *StFH* genes were involved in biological processes, with 63 out of 74 GO terms affecting actin polymerization. Six major transcription factor families, including bZIP, C2H2, ERF, GATA, LBD, NAC, and HSF, were identified that were involved in the regulation of *StFH* genes in various abiotic stresses, including drought. Further, the 60 unique microRNAs targeted 24 *StFH* by regulating gene expression in response to drought stress were identified. The expression of *StFH* genes in 14 different tissues, particularly in drought-responsive tissues such as root, stem, shoot apex, and leaf, underscores their significance in managing drought stress. RNA-seq analysis of the drought-resistant Qingshu No. 9 variety revealed the potential role of up-regulated genes, including *StFH2*, *StFH10*, *StFH19*, and *StFH25*, in

**Funding:** The author(s) received no specific funding for this work.

**Competing interests:** The authors have declared that no competing interests exist.

alleviating drought stress. Overall, these findings provide crucial insights into the response to drought stress in potatoes and can be utilized in breeding programs to develop potato cultivars with enhanced drought-tolerant traits.

## 1.0 Introduction

Formins, a variety of multi-domain proteins, efficiently stimulate actin polymerization in living organisms, indicating their role as actin nucleators [1]. Formin proteins are comprised of multiple domains, prominently featuring FH1 (Formin Homology 1), FH2 (Formin Homology 2), and FH3 (Formin Homology 3) [2]. The FH1 domain contains polyproline tracks that bind to profilin [3]. Profilin transports actin subunits to the FH2 domain [4]. Additionally, profilin enhances the elongation rate of actin filaments associated with formins, often surpassing the speed of free actin filaments [5]. Plants have developed advanced strategies to deal with environmental stresses. The cytoskeleton, a dynamic network of actin filaments and microtubules within cells, is an integral aspect of this adaptation [6]. This actin filament structure plays a crucial role in facilitating rapid changes in the shape and structure of cells, enabling plants to adapt their organs and tissues for improved stress resilience [7]. Notably, cytoskeleton dynamics contribute to salt tolerance in *Arabidopsis thaliana* and enhance drought resistance in different plant species [8]. Although the precise mechanisms are still under investigation, the cytoskeleton plays a fundamental role in the plant stress response [9].

Plants generally rely on the cytoskeleton's dynamic adaptation to various environmental stressors [10]. When stress challenges the cell, the cytoskeleton reorganizes its components, reinforcing weak points to ensure resilience [11]. Specialized proteins such as tubulins and actins are crucial in the cytoskeleton's stress response [12]. These proteins maintain structural integrity, facilitate movements, and support cellular communication for collective stress response [8]. Another aspect of the cell's internal structure, microtubules (MT), act as the cell's internal support system. When the cell faces stress, especially from salt, these microtubules help by both building up and breaking down. Signal transduction mechanisms such as abscisic acid (ABA) regulation, cytosolic calcium ions, and reactive oxygen species (ROS) are crucial in controlling stress in plants. High levels of cytosolic calcium ions and ABA in the root ensure that the structure of cortical MTs breaks down to depolymerize. Conversely, ROS in plants rebuilds these microtubules by rearranging α-tubulin and β-tubulin to encounter the drought stress conditions [8, 13, 14].

Potato (*Solanum tuberosum* L.), commonly known as white potato or Irish potato, is cultivated on 19.3 million hectares globally and yields almost 400 million metric tons annually, ranking only behind rice, wheat, and maize [15]. Stress-tolerant potato cultivars are necessary to maintain the food supply despite climate change and protect global populations [16]. Although potatoes are susceptible to common stressors such as drought and late blight, leading to significant reductions in yield, they demonstrate inherent adaptability through several stress-response mechanisms, such as salinity adaptation and antioxidant production. The formin gene family members have been identified in various economically valuable plant species, such as 25 and 34 in wheat (*Triticum aestivum* L.) [17] and soybean (*Glycine max*. L.) [18], respectively. However, formin genes have not been investigated in the genome of potato (*S. tuberosum* L.).

This study characterized 26 *StFH* genes, revealing their physical and chemical properties, phylogenetic comparison, genomic evolution, gene structure, conserved domain, motifs, gene

duplication, chromosome mapping, subcellular localization, *cis*-acting regulatory elements, tissue-specific expression, and RNA-seq data analysis under drought stress condition. The findings from this study will build the basis for functional investigations on the *StFH* genes and offer excellent opportunities to improve this crop species with drought-tolerant traits in future breeding programs.

## 2.0 Materials and methods

### 2.1 Database searching and retrieval of formin proteins in the potato genome

The *A. thaliana* formin DNA-binding domains were used to retrieve gene-encoded formin proteins in the potato (*S. tuberosum*) genome using the BlastP version 2.15.0 (Protein-basic local alignment search tool) program at Phytozome v13 (https://phytozome-next.jgi.doe.gov/) [19] (S1 Data). The comparison matrix (BLOSUM62), an expected (E) threshold value of -1, and other default parameters were employed. The Pfam database as Hidden Markov Models (HMMER) [20], Simple Modular Architecture Research Tool (SMART) (http://smart.embl-heidelberg.de/) [21], and NCBI Conserved Domain Database (CDD) (https://www.ncbi.nlm.nih.gov/Structure/cdd/wrpsb.cgi) [22] were used to analyze the presence of the FH2 conserved domain with default parameters. StFH proteins containing FH2 domains were renamed according to their chromosomal positions.

### 2.2 Determination of physio-chemical properties for StFH

The physiochemical characteristics of StFH proteins, including the number of amino acids (A.A.), molecular weight, isoelectric points (pI), and grand average hydrophilicity (GRAVY), were determined in the ProtParam online program (http://web.expasy.org/protparam/) [23].

### 2.3 Phylogenetic relationship analysis between ZmFH, OsFH, AtFH, MtFH, LjFH, and StFH

The evolutionary phylogenetic tree of FH2 proteins between *Zea mays*, *Oryza sativa*, *Arabidopsis thaliana*, *Medicago truncatula*, *Lotus japonicas*, and *Solanum tuberosum* was constructed (S2 Data). The sequences were aligned using the MEGA version 11.0.10 software (https://www.megasoftware.net/) [24]. The ClustalW [25] program was employed to align amino acid sequences. The Maximum Likelihood (ML) method was used with default parameters and a significant 1000 bootstrap value to strengthen branch values [26]. The FASTA format of the phylogenetic tree was further processed using IQ Tree version 1.6.12 with default parameters (http://www.iqtree.org/) [27]. The finalized phylogenetic tree was uploaded to iTOL v6.7.4 (https://itol.embl.de/) for an attractive visual representation [28].

### 2.4 Gene structure analysis of *StFH*

CDS (coding sequence) and genomic DNA sequences were extracted in the FASTA format from Phytozome v13.0 to elucidate the structure of the *StFH* genes using the Gene Structure Display Server (GSDS v2.0) (http://gsds.cbi.pku.edu.cn/) [29] (S3 and S4 Data).

### 2.5 Conserved domain and motif analysis of StFH

The DOG2.0 software was used to demonstrate the conserved domain of StFH [30]. The specific position details of the FH2 domain sequence were collected through MOTIF-searching protein sequence motifs (https://www.genome.jp/tools/motif/) [31]. To analyze the structural

motifs of the StFH, Multiple EM for Motif Elicitation (MEME) tools (https://meme-suite.org/meme/tools/meme) was employed with a maximum of 20 motifs and other default parameters [32]. The structural pattern of motifs was visualized in TB tools v.2.010 [33].

## 2.6 The evolutionary Ka/Ks ratio analysis of *StFH*

The Ka (non-synonymous) and Ks (synonymous) substitution ratios of *StFH* were computed in the Ka/Ks calculation tool (https://services.cbu.uib.no/tools/kaks) [34]. Specifically, the MCScanX tool of TB tools v.2.010 was employed on the *StFH* CDS of duplicated genes. The Ka/Ks ratio was used to determine the rates of molecular evolution for each pair of paralogous genes, providing insights into the selective pressures acting on these duplicated sequences. Additionally, the duplication and time of divergence for the *StFH* genes were measured in million years ago (MYA) using the formula $T = Ks/2x$, where x was equal to $6.56 \times 10^{-9}$ and $MYA = 10^{-6}$ [35].

## 2.7 Collinearity and synteny analysis of *StFH*

The collinear relationships of *StFH* genes were analyzed based on gene duplication events. Additionally, the syntenic relationships between *S. tuberosum* and *A. thaliana*, *Z. mays*, and *O. sativa* were analyzed using MCScanX in TB tools v.2.010. Both analyses were visualized with TB tools v.2.010.

## 2.8 Chromosomal localization and duplication analysis of *StFH*

The details on chromosomes were extracted from Phytozome v13, and the distribution of *StFH* genes throughout the chromosomes was mapped using the MapGene2Chrom web v2 (MG2C) web server (http://mg2c.iask.in/mg2c_v2.0/) [36]. Segmental and tandem duplicated gene pairs were analyzed in the distributed chromosomes.

## 2.9 Distribution of *StFH* on multiple sub-genomes of *S. tuberosum*

The chromosomal lengths, as well as the start, and end points for the three sub-genomes of *S. tuberosum*, including *S. tuberosum* cv. Otava (autotetraploid), *S. tuberosum* group Phureja DM 1–3516 R44 (doubled monoploid), and *S. tuberosum* group Tuberosum RH89-039-16 (heterozygous diploid), were obtained from the SpudDB database version 6.1 (http://spuddb.uga.edu/) [37]. The distribution of *StFH* genes in these sub-genomes was visualized using the MapGene2Chrom web v2 (MG2C) web server.

## 2.10 Subcellular localization analysis of *StFH*

The subcellular localization of *StFH* was analyzed in the Wolf PSORT online tool (https://wolfpsort.hgc.jp/) [38]. The predicted results were visualized using the RStudio 2023.06.1 version [39].

## 2.11 *Cis*-acting regulatory elements (CAREs) analysis of *StFH*

The 2000 bp from the 5′ untranslated region (5′ UTR) of the *StFH* gene was collected from the Phytozome v13 (S5 Data). The predictions of CAREs were conducted in the PlantCARE database (http://bioinformatics.psb.ugent.be/webtools/plantcare/html/) [40]. The predicted CAREs were further classified and visualized as a heatmap in TB tools v.2.010.

## 2.12 Gene ontology (GO) analysis of *StFH*

GO analysis was performed by obtaining GO IDs from the Plant Transcriptional Regulatory Map database (PlantRegMap; https://plantregmap.gao-lab.org/go.php) using the *p*-value of 0.01 and other default parameters [41]. The ChiPlot online tool (https://www.chiplot.online/) was used to illustrate the categorization and functions of *StFH* genes [42].

## 2.13 Transcription factors (TF) analysis of *StFH*

The analysis of TF was conducted in the PlantRegMap database (https://plantregmap.gao-lab.org/binding_site_prediction.php) using the threshold *p*-value 1.0E-4 and other default parameters. RStudio version 2023.06.1 was used to visualize and categorize the TFs family.

## 2.14 Regulatory network between TFs and *StFH*

Cytoscape v3.10.0 was used to visualize interactions between major TF families and respective *StFH* genes [43].

## 2.15 Prediction of putative micro-RNAs (miRNAs) and networks targeting *StFH*

Potential regions targeted by miRNAs in *StFH* genes were identified using miRNA sequences of *S. tuberosum* from miRBase (https://mirbase.org/) [44]. The CDS of *StFH* genes were uploaded to the psRNATarget Server 18 (https://www.zhaolab.org/psRNATarget/analysis?function=2), using default parameters to predict potential miRNA interactions with *StFH* genes [45]. Cytoscape version 3.10.0 was used to construct and visualize the interaction network between the predicted miRNAs and the *StFH* genes targeted by miRNA.

## 2.16 Protein-protein interaction (PPI) of StFH

The web tool string version 12 (https://string-db.org/) was utilized to predict and construct the PPI network of StFH proteins based on homologous proteins in *A. thaliana* [46]. The parameters were as follows: network type-full STRING network, network edges evidence-meaning, active interaction source-text mining, experiments, databases, co-expression, neighborhood, gene fusion, and co-occurrence. The minimum required interaction score was defined as a medium confidence parameter (0.4). For the first shell, a maximum of 10 interactions were displayed, and the second shell was left blank. Network display options were enabled with a 3D bubble design.

## 2.17 Tissue-specific expression of *StFH*

The SpudDB database was used to extract the tissue-specific expression of the *S. tuberosum* group, Tuberosum RH89-039-16. The data consisted of 14 different tissues and were classified into four groups; floral (stamen and flower), leaf (leaf, petiole), stolon/tuber (stolon, tuber sprout, tuber peel, tuber pith, tuber cortex, young tuber, and mature tuber), and other tissues (stem, shoot apex, and root) [47]. Consequently, the PPM (parts per million) value greater than >0.2 is considered as an expressed gene. TB tools v.2.010 was used to illustrate the expression patterns.

## 2.18 Expression pattern of *StFH* under drought stress

The RNA-seq data was composed of two potato varieties; Atlantic (a drought-sensitive variety) and Qingshu No. 9 (a drought-resistant variety). The *StFH* genes were analyzed under drought

stress in early flowering, full blooming, and flower falling states at three different time points (25, 50, and 75 days, respectively). The sequence read archive (SRA) at NCBI under the bio-project ID PRJNA541096 was extracted [48]. The trimmomatic package version 0.32 was utilized for quality control and trimming of the RNA-seq data [49]. Besides, RNA-seq data was aligned to the *S. tuberosum* reference genome using the STAR package version 2.7.11b [50]. Conversion of sequence alignment map (SAM) files to binary alignment map (BAM) files, sorting, and arrangement were performed using the samtools version 1.20 packages [51]. The fragments per kilobase million (FPKM) value was calculated using the RSEM package v1.1.17 [52]. The FPKM value greater than >0.2 was considered as an expressed gene. TB tools v.2.010 was employed for generating a heatmap to visually represent the expression patterns of *StFH* under drought stress.

## 3.0 Results

### 3.1 Determination of physio-chemical properties of StFH proteins

The physio-chemical properties of StFH indicated a significant variability in A.A., ranging from 79 A.A. (StFH14) to 1746 A.A. (StFH8), with an average length of 912.5 A.A. (Table 1). The encoded StFH proteins showed diverse molecular weights, ranging from 8881.02 kDa (StFH14) to 186282.23 kDa (StFH8). The pI measurements indicated a distinct acidic nature in 13 StFHs (StFH8, StFH9, StFH13, StFH14, StFH17, StFH19, StFH20, StFH21, StFH22,

**Table 1. List of 26 StFH proteins and their basic physio-chemical characterization.**

| Gene name | Gene identifier | Size (A.A.) | Mass (kDa) | pI | Instability index | Aliphatic index | Grand average of hydropathicity (GRAVY) |
|-----------|-----------------|-------------|------------|-----|-------------------|-----------------|------------------------------------------|
| *StFH1* | Soltu.DM.01G039360 | 900 | 99423.69 | 8.02 | 57.12 | 84.46 | -0.343 |
| *StFH2* | Soltu.DM.02G027760 | 886 | 97885.23 | 9.11 | 55.73 | 75.15 | -0.606 |
| *StFH3* | Soltu.DM.03G005470 | 881 | 97837.35 | 9.26 | 51.69 | 77.80 | -0.546 |
| *StFH4* | Soltu.DM.03G027630 | 755 | 84258.78 | 8.13 | 53.26 | 75.92 | -0.583 |
| *StFH5* | Soltu.DM.03G029790 | 1523 | 164989.25 | 7.94 | 66.52 | 66.79 | -0.556 |
| *StFH6* | Soltu.DM.05G007060 | 800 | 87394.25 | 9.33 | 46.95 | 80.71 | -0.386 |
| *StFH7* | Soltu.DM.05G018890 | 923 | 100495.71 | 8.57 | 45.06 | 80.92 | -0.453 |
| *StFH8* | Soltu.DM.06G025740 | 1746 | 186282.23 | 6.88 | 78.20 | 68.14 | -0.538 |
| *StFH9* | Soltu.DM.06G033180 | 944 | 103548.17 | 6.41 | 68.18 | 74.87 | -0.449 |
| *StFH10* | Soltu.DM.07G000540 | 1470 | 160849.62 | 7.64 | 63.22 | 77.34 | -0.447 |
| *StFH11* | Soltu.DM.07G018190 | 931 | 102161.23 | 8.95 | 60.06 | 79.88 | -0.441 |
| *StFH12* | Soltu.DM.07G018220 | 107 | 12557.53 | 7.66 | 56.44 | 89.35 | -0.328 |
| *StFH13* | Soltu.DM.07G018270 | 91 | 10140.41 | 4.64 | 37.20 | 107.03 | -0.226 |
| *StFH14* | Soltu.DM.07G018320 | 79 | 8881.02 | 4.21 | 37.71 | 97.47 | -0.249 |
| *StFH15* | Soltu.DM.07G018330 | 210 | 24194.69 | 8.20 | 52.41 | 86.38 | -0.280 |
| *StFH16* | Soltu.DM.07G025400 | 880 | 96597.43 | 8.79 | 50.09 | 80.85 | -0.487 |
| *StFH17* | Soltu.DM.08G006050 | 755 | 84664.51 | 6.22 | 55.20 | 86.54 | -0.296 |
| *StFH18* | Soltu.DM.08G021110 | 1329 | 144466.43 | 8.17 | 66.32 | 75.38 | -0.458 |
| *StFH19* | Soltu.DM.09G015150 | 983 | 107851.80 | 6.93 | 74.05 | 80.53 | -0.393 |
| *StFH20* | Soltu.DM.10G001740 | 920 | 100485.36 | 6.46 | 48.47 | 79.75 | -0.432 |
| *StFH21* | Soltu.DM.10G001770 | 50 | 54435.60 | 6.44 | 35.87 | 84.12 | -0.378 |
| *StFH22* | Soltu.DM.10G001820 | 150 | 16612.90 | 6.74 | 41.54 | 81.33 | -0.538 |
| *StFH23* | Soltu.DM.10G001850 | 382 | 42384.39 | 5.06 | 33.12 | 87.09 | -0.318 |
| *StFH24* | Soltu.DM.10G001880 | 402 | 44744.22 | 5.38 | 51.21 | 86.37 | -0.301 |
| *StFH25* | Soltu.DM.12G019570 | 944 | 103983.11 | 6.07 | 59.23 | 80.95 | -0.355 |
| *StFH26* | Soltu.DM.12G027420 | 1324 | 147544.15 | 6.27 | 59.37 | 84.58 | -0.387 |

StFH23, StFH24, StFH25, and StFH26) with pI<7.0, whereas 13 others (StFH1, StFH2, StFH3, StFH4, StFH5, StFH6, StFH7, StFH10, StFH11, StFH12, StFH15, StFH16, and StFH18) exhibited an alkaline nature with pI>7.0. Moreover, 22 StFH proteins (StFH1, StFH2, StFH3, StFH4, StFH5, StFH6, StFH7, StFH8, StFH9, StFH10, StFH11, StFH12, StFH15, StFH16, StFH17, StFH18, StFH19, StFH20, StFH22, StFH24, StFH25, StFH26) with values exceeding instability index 40, indicating potential structural instability, whereas four proteins (StFH13, StFH14, StFH21, StFH23) exhibited lower instability index values less than 40.0. The aliphatic index exhibited significant diversity among the *StFH* genes, with StFH13 having the highest value of 107.03 and StFH26 having the lowest index of 66.79. The hydrophilic nature of all the StFH proteins was confirmed by the consistently negative GRAVY scores.

## 3.2 Phylogenetic relationship analysis between ZmFH, OsFH, AtFH, MtFH, LjFH, and StFH

The phylogenetic relationship analysis included a total of 117 formin proteins; 26 StFH, 21 AtFH, 20 ZmFH, 19 MtFH, 17 OsFH, and 14 LjFH, which were categorized into seven distinct groups: group A (19 proteins), group B (9 proteins), group C (14 proteins), group D (31 proteins), group E (7 proteins), group F (15 proteins), and group G (22 proteins) (Fig 1). The distribution of StFH proteins within these groups was as follows: group A (3 proteins)-StFH5, StFH8 and StFH18, group B (2 proteins)-StFH10, and StFH26, group C (2 proteins)-StFH4 and StFH6, group D (12 proteins)-StFH7, StFH11, STFH12, StFH13, StFH14, StFH15, StFH16, StFH20, StFH21, StFH22, StFH23, and StFH24, group E (1 protein)-StFH1, group F (2 proteins)-StFH2 and StFH3, and group G (4 proteins)-StFH9, StFH17, StFH19, and StFH25 (S6 Data).

## 3.3 Gene structure analysis of *StFH*

The structural diversity of *StFH* genes was analyzed by comparing introns and exons based on their distribution patterns. The group A members (*StFH5*, *StFH8*, and *StFH18*), each containing 17 exons, shared similar structural patterns (Fig 2 and S7 Data). Meanwhile, group B members (*StFH10* and *StFH26*) exhibited structural similarity with group A, with *StFH26* having the highest numbers of exons and introns (19 and 18, respectively). The group C members (*StFH4* and *StFH6*) exhibited the lowest numbers of exons and introns but had similar structural patterns. Whereas, group D, containing 12 genes, exhibited the highest structural complexity with a total of 59 exons and 52 introns. But group E contained only *StFH1*, with four exons and three introns. Besides, group F exhibited distinct structures, with *StFH2* having four exons and three introns, while *StFH3* has five exons and four introns. Similarly, group G members (*StFH9*, *StrFH17*, *StFH19*, and *StFH25*) shared a consistent pattern of four exons and three introns.

## 3.4 Conserved domain and motif analysis of StFH

The analysis of conserved domains revealed the presence of the FH2 domain across all 26 StFH proteins (Fig 3). The motif analysis uncovered unique patterns across different StFH groups. The groups C (StFH4), G (StFH9, StFH19, and StFH25), and F (StFH3) each contained 11 motifs (Fig 4). While groups F (StFH2) and D (StFH11) both contained 12 motifs. Moreover, groups A members (StFH8, and StFH18) and B members (StFH10) each contained 14 motifs. StFH5 (group A) had 15 motifs, while StFH26 (group B) had 13 motifs. However, StFH1, the only member of group E, contained 13 motifs. Furthermore, other group D members displayed a variety of motifs, indicating structural diversity and genomic differences among them.

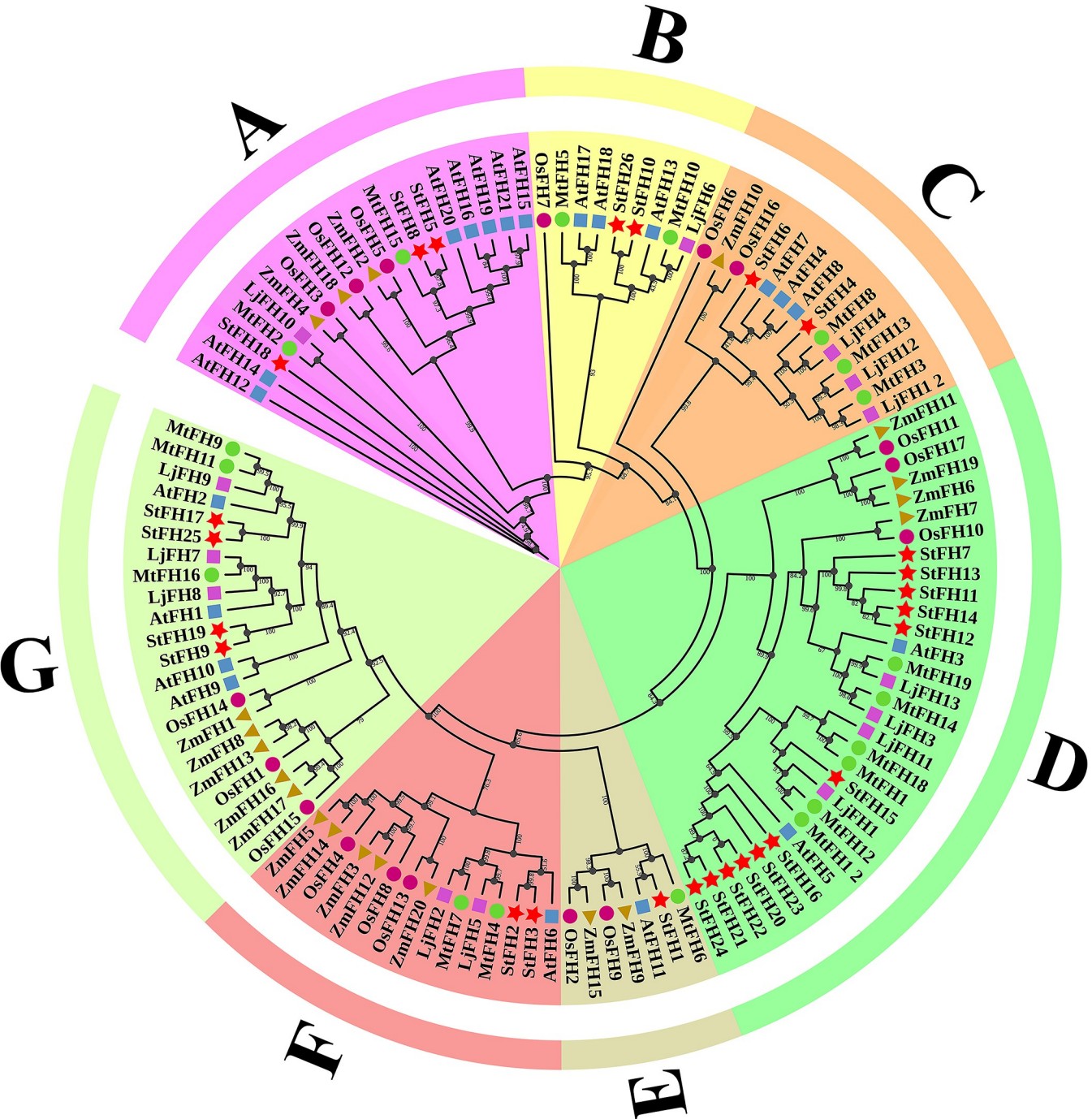

**Fig 1. Phylogenetic relationship between StFH, AtFH, ZmFH, MtFH, LjFH, and OsFH.** StFHs were classified into seven groups (A, B, C, D, E, F, and G). Different colors and shapes mark each gene family, with StFH labeled with red stars, AtFH with rectangular blue rectangles, ZmFH with dark yellow triangles, MtFH with green ellipses, OsFH with red-violet circles, and LjFH with purple rectangles.

### 3.5 The evolutionary Ka/Ks ratio analysis of *StFH*

The estimated Ka values for the *StFH* genes ranged from 0.147052693 to 0.219661933, while the Ks values varied from 0.248696445 to 1.13504586 (Fig 5 and S8 Data). None of the *StFH*

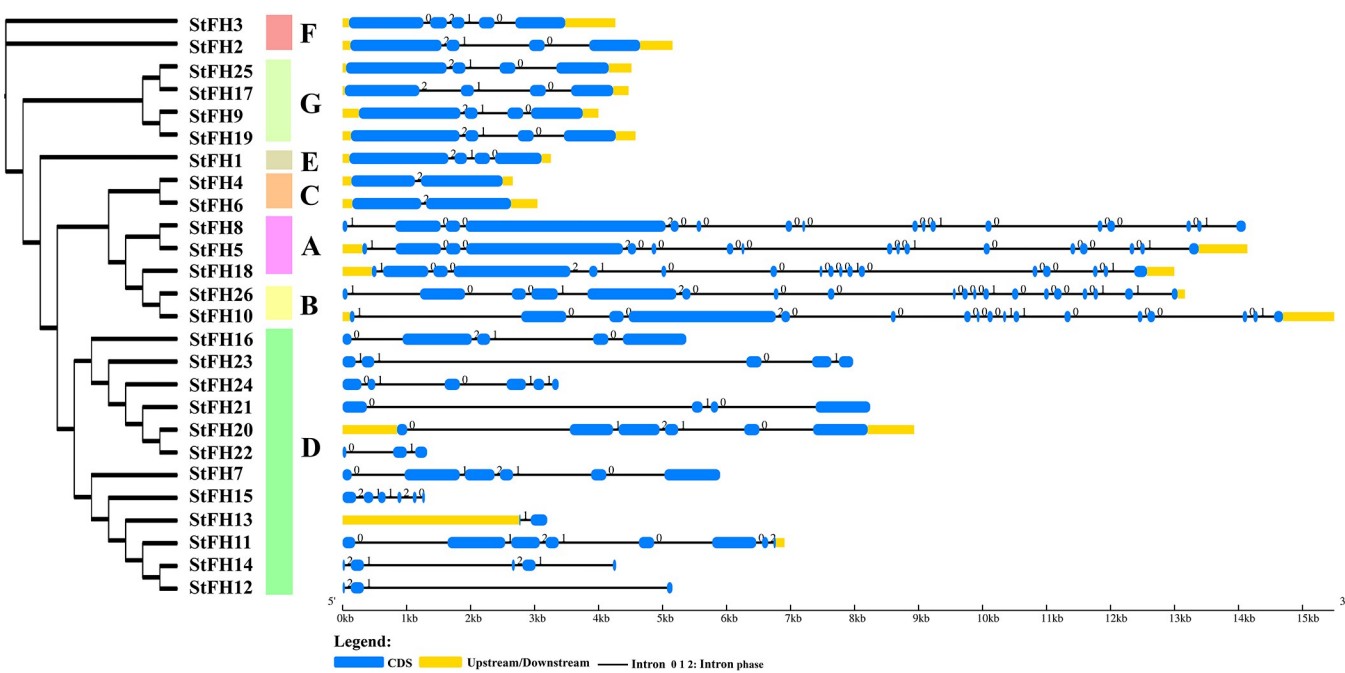

**Fig 2. The structure of *StFH* genes.** The grouping and colors of the *StFH* members were based on the phylogenetic tree. Black lines represent introns, deep blue represents exons, numbers 0, 1, and 2 represent intron phases, and deep yellow lines represent upstream/downstream.

gene pairs were undergoing positive selection as their Ka/Ks ratio is less than 1. Gene pairs such as *StFH5-StFH8* (0.219168631), *StFH26-StFH10* (0.3238209), *StFH2-StFH3* (0.234076034), *StFH25-StFH17* (0.193526923), *StFH9-StFH19* (0.25279716), *StFH21-StFH24* (0.591293908), and *StFH14-StFH12* (0.661554621) were all undergoing purifying selection.

### 3.6 Collinearity relationship analysis of *StFH*

A close relationship among the *StFH* genes was demonstrated through collinear analysis (Fig 6). Seven collinear pairs were identified within the *StFH* gene family. Chromosome 3 of *StFH5* formed a pair with chromosome 6 of *StFH8*. Chromosome 2 of the *StFH2* interacted with chromosome 3 of the *StFH3*. *StFH10*, located on chromosome 7, is paired with *StFH26* on chromosome 12. *StFH17* on chromosome 8 formed a collinear pair with *StFH25* on chromosome 12. *StFH9* on chromosome 6 interacted with *StFH19* on chromosome 9. Additionally, *StFH12* and *StFH14* formed a pair on chromosome 7, and *StFH21* and *StFH24* were also observed forming collinear pairs on chromosome 10.

### 3.7 Synteny relationship analysis of *StFH*

The syntenic comparison of *S. tuberosum* (dicotyledon) was further examined with two monocotyledons, *Z. mays* and *O. sativa*, and one dicotyledon, *A. thaliana* (Fig 7). The results uncovered the close syntenic association of *S. tuberosum* (12 chromosomes) with *A. thaliana* (5 chromosomes). They shared 11 gene pairs on distinct chromosomes, indicating similarity in biological and physiological functions. On the contrary, among the monocotyledon species, *S. tuberosum* showed more similarity in evolutionary relations with 7 gene pairs with *Z. mays* (9 chromosomes) compared to 5 gene pairs with *O. sativa* (12 chromosomes). Thus, it contributed to understanding ancestry and genomic relations.

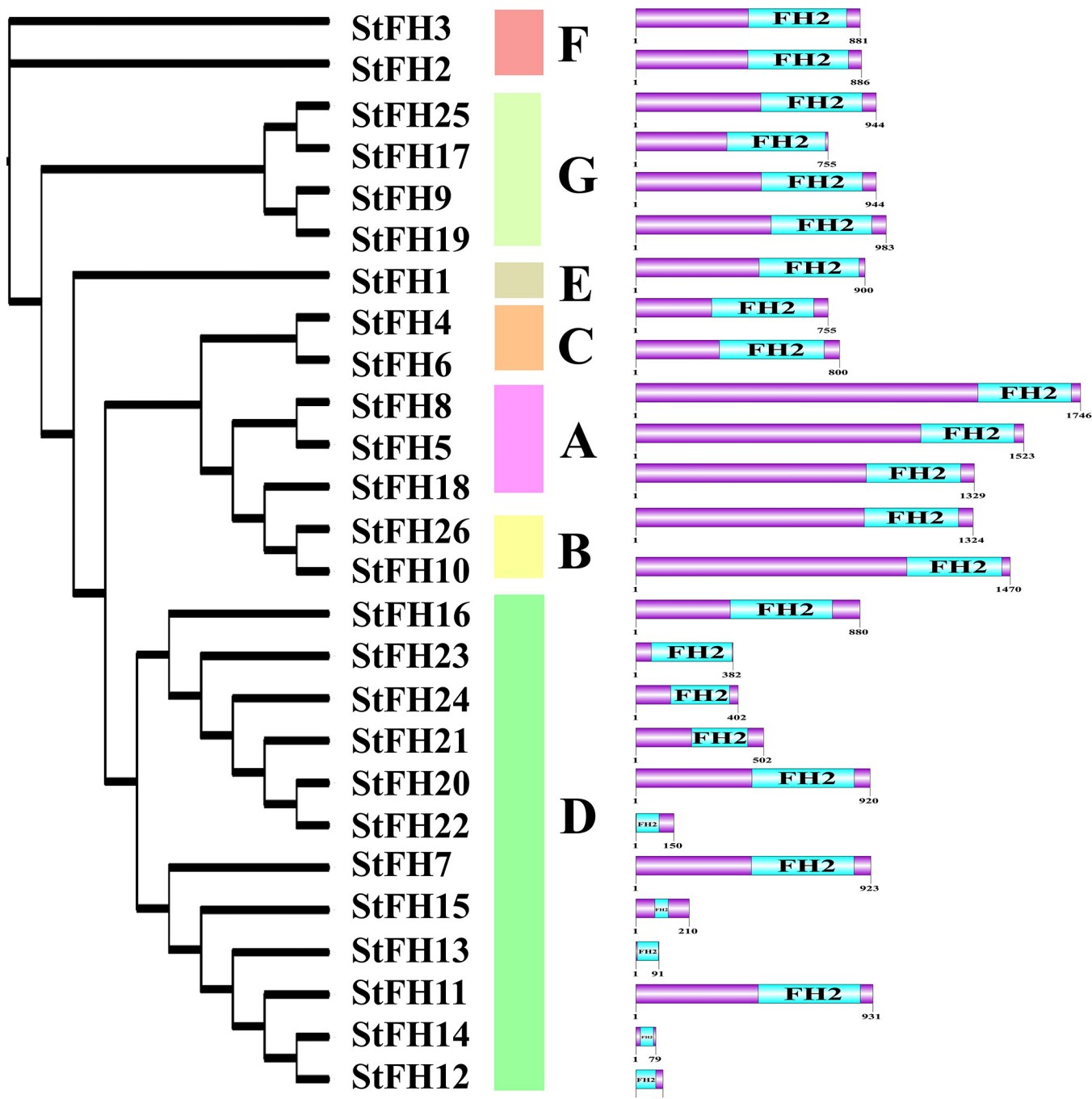

**Fig 3. Feature domains of StFH.** Positions of the FH2 conserved domain and other domains are demonstrated in light blue color, whereas the entire protein sequence of the respective StFH is shown in purple.

### 3.8 Chromosomal localization and duplication analysis of *StFH*

The 26 *StFH* genes were distributed on ten chromosomes (Fig 8). Single *StFH* genes (*StFH1*, *StFH2*, and *StFH19*) were located on chromosomes 1, 2, and 9, respectively. The remaining 23 *StFH* genes were located on chromosome 3 (*StFH3*, *StFH4*, and *StFH5*), chromosome 5 (*StFH6* and *StFH7*), chromosome 6 (*StFH8* and *StFH9*), chromosome 7 (*StFH10*, *StFH11*,

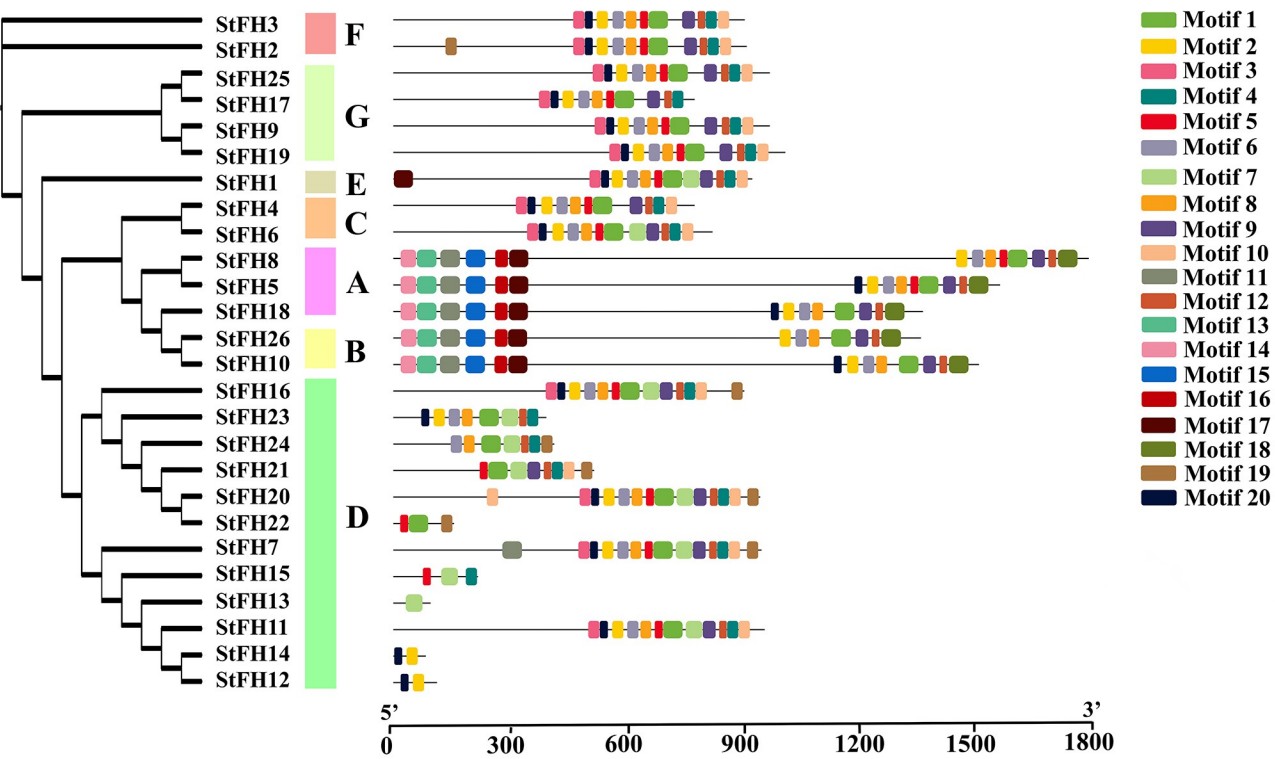

**Fig 4. The distribution of conserved motifs in StFH.** Each motif was illustrated by a color box aligned on the right side of the figure. Different colors indicate individual motifs within each StFH protein.

*StFH12, StFH13, StFH14, StFH15,* and *StFH16),* chromosome 8 *(StFH17* and *StFH18),* chromosome 10 *(StFH20, StFH21, StFH23,* and *StFH24),* and chromosome 12 *(StFH25* and *StFH26).* However, chromosomes 4 and 11 did not contain any *StFH* genes. Meanwhile, two tandem and five segmental duplications were identified. The five identified segmental pairs were; *StFH8-StFH5, StFH26-StFH10, StFH2-StFH3, StFH25-StFH17,* and *StFH9-StFH19.* The two tandem pairs were *StFH21-StFH24* and *StFH12-StFH14.*

### 3.9 Distribution of *StFH* on multiple sub-genomes of *S. tuberosum*

The StFH genes were distributed randomly across different chromosomes in *S. tuberosum* cv. Otava (autotetraploid), cv. Phureja DM 1–3516 R44 (doubled monoploid), and cv. Tuberosum RH89-039-16 (heterozygous diploid) (Fig 9). The cultivar Otava contained 22 *StFH* in 10 chromosomes, with chromosome 7 containing the highest number (8) of genes. Meanwhile, cultivar Phureja DM 1–3516 R44 had 23 *StFH* in 9 chromosomes, with a maximum number of 6 genes located in chromosome 7. The cultivar Tuberosum RH89-039-16 comprised 40 *StFH* distributed in 18 chromosomes. However, most of the three sub-genomic chromosomes in tuberosum contained one or two genes.

### 3.10 Prediction of the subcellular localization of StFH

The subcellular localization analysis revealed that StFH protein signals were present in 11 different organelles, including the nucleus, mitochondria, chloroplast, cytoplasm, cytoskeleton, Golgi apparatus, peroxisome, vacuole, endoplasmic reticulum (ER), plasma membrane (PM), and extracellular membranes. StFH9 signals were detected at the highest number of sites,

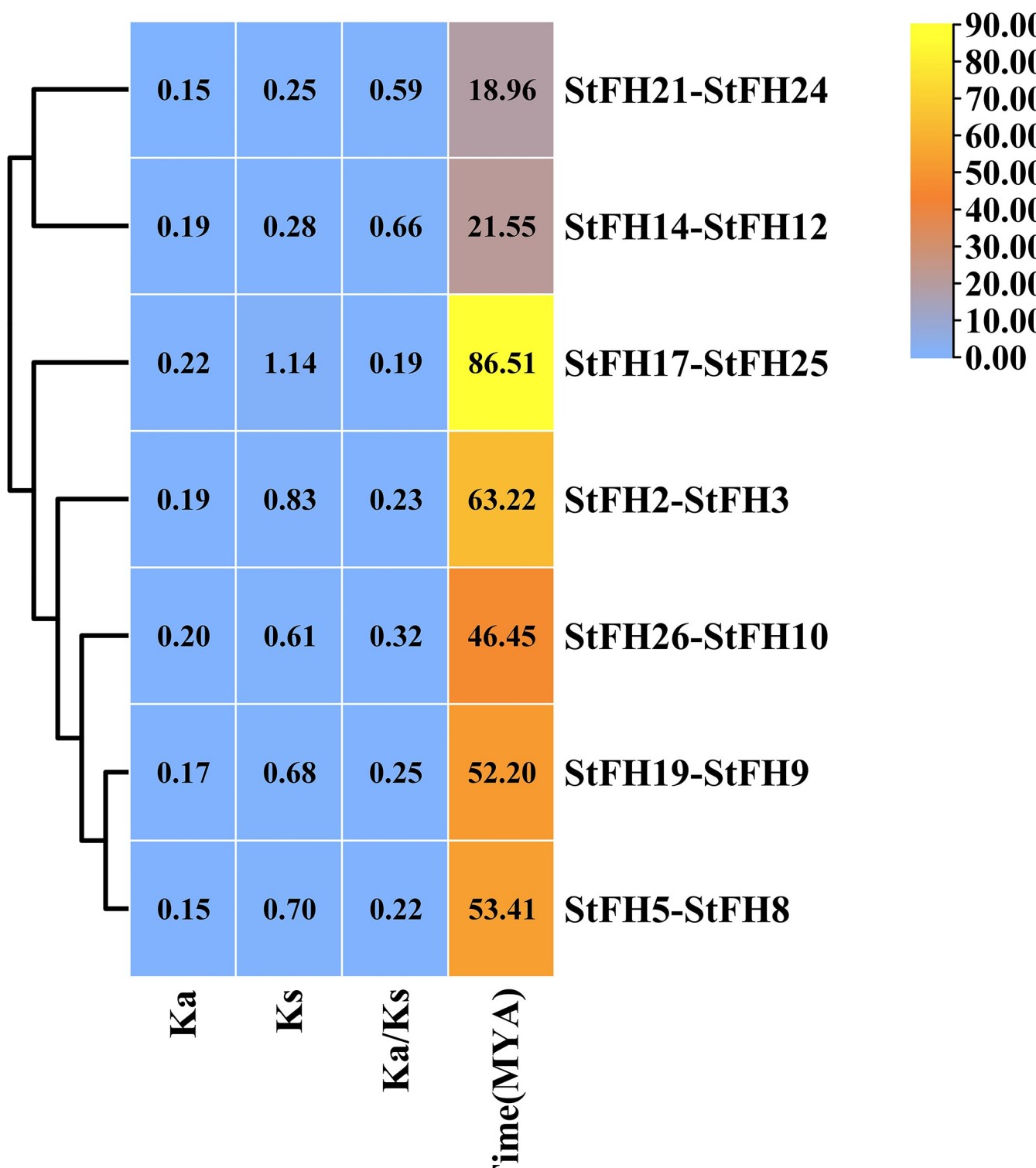

**Fig 5. Estimation of gene duplication time and Ka/Ks analysis of *StFH*.** The ratio of Ka to Ks is represented by Ka/Ks, with divergence time (measured in million years ago, MYA) also indicated. The color bar represents the data range.

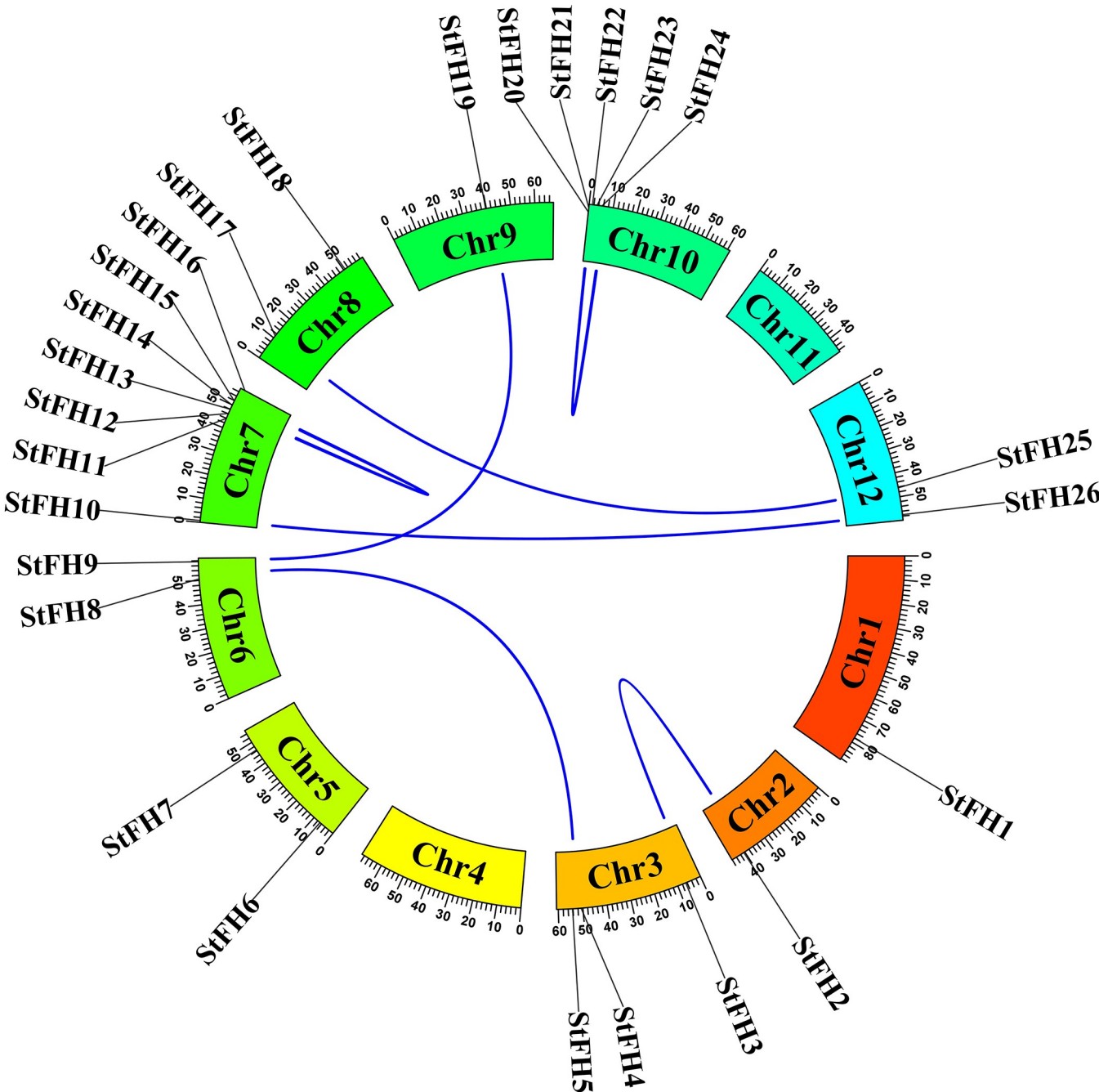

**Fig 6. The collinearity analysis of the *StFH* gene family.** Different color rectangles represent chromosomes 1 to 12 in *StFH*. The dark blue lines linking chromosomes represent collinear relationships between them.

including all 11 sites of chloroplasts (Fig 10A). The protein signals were most abundant in the chloroplast (69.23%) and plasma membrane (57.69%), followed by the vacuole (53.84%) and nucleus (50%). Significant quantities of StFH protein signals were also found in the ER (46.15%), cytoplasm (42.3%), mitochondria (38.46%), extracellular cytoskeleton (30.76%), and Golgi body (26.92%). The peroxisome had the lowest concentration (3.84%) of StFH signals

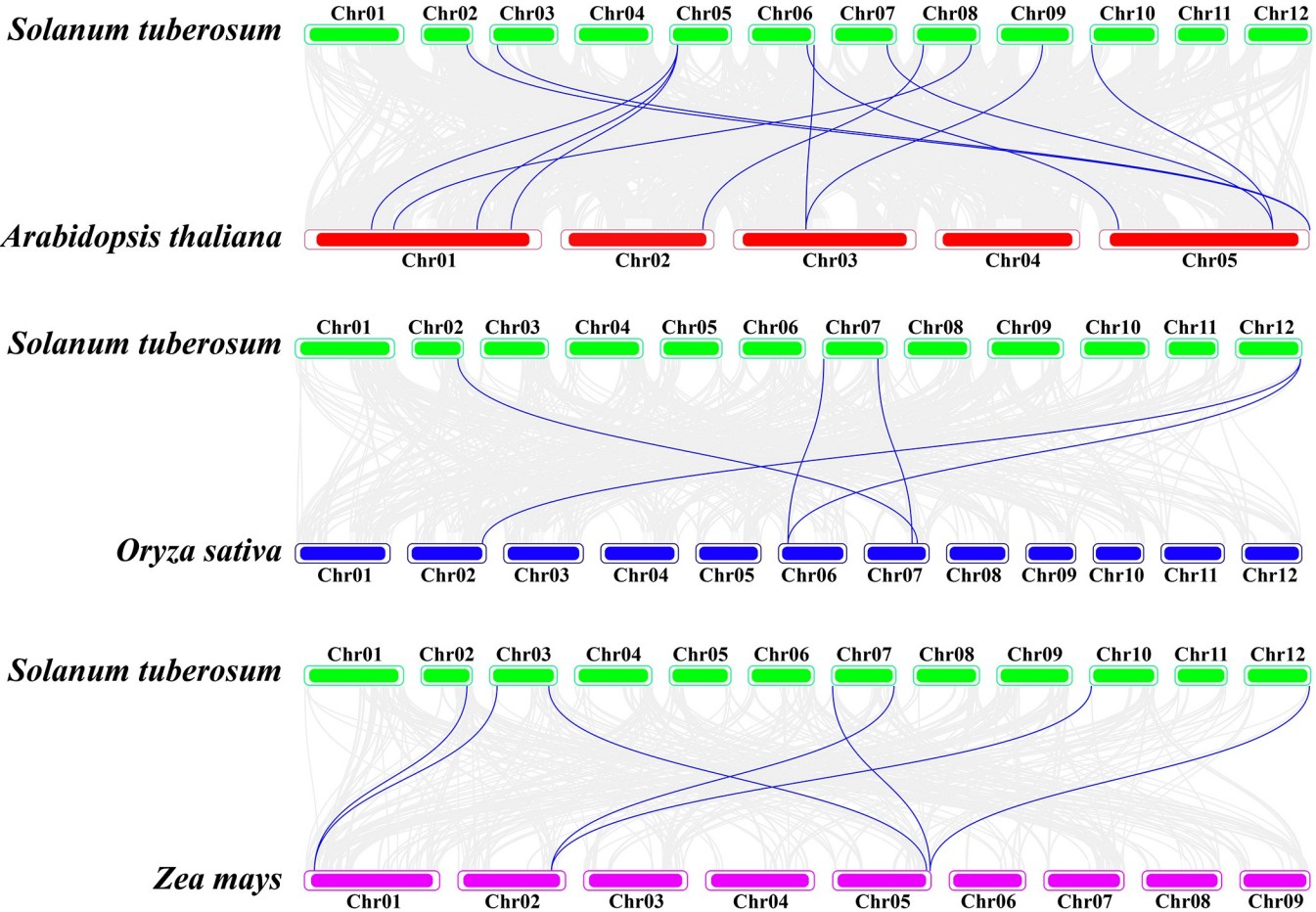

**Fig 7. The synteny analysis of *S. tuberosum* with *A. thaliana*, *O. sativa*, and *Z. mays*.** The green, red, blue, and magenta-colored chromosome rectangles represent *S. tuberosum*, *A. thaliana*, *O. sativa*, and *Z. mays*. The dark blue color indicates a syntenic relationship between the species.

(Fig 10B). The bubble plot illustrated the redundancy of a particular StFH gene in specific organelles (S1 Fig).

### 3.11 *Cis*-acting regulatory elements (CAREs) analysis in the promoters of *StFH*

The analysis of CAREs revealed 54 elements, with the most abundant category being Box 4, related to light responsiveness. CAREs were classified into four categories based on their functional regulation: light responsiveness, tissue-specific expression, phytohormone responsiveness, and stress responsiveness (Fig 11 and S9 Data). The largest category of CAREs related to light responsiveness included 26 elements such as 3-AF1 binding site, AAAC-motif, ACA-motif, ACE, AE-box, AT1-motif, ATC-motif, ATCT-motif, Box 4, Box II, CAG-motif, chs-CMA1a, chs-CMA2a, GA-motif, Gap-box, GATA-motif, G-Box, GT1-motif, GTGGC-motif, I-box, LAMP-element, MRE, Sp1, TCCC-motif, and TCT-motif. The tissue-specific expression CAREs included 13 elements, phytohormone responsiveness included 12 elements, and stress responsiveness included 4 elements.

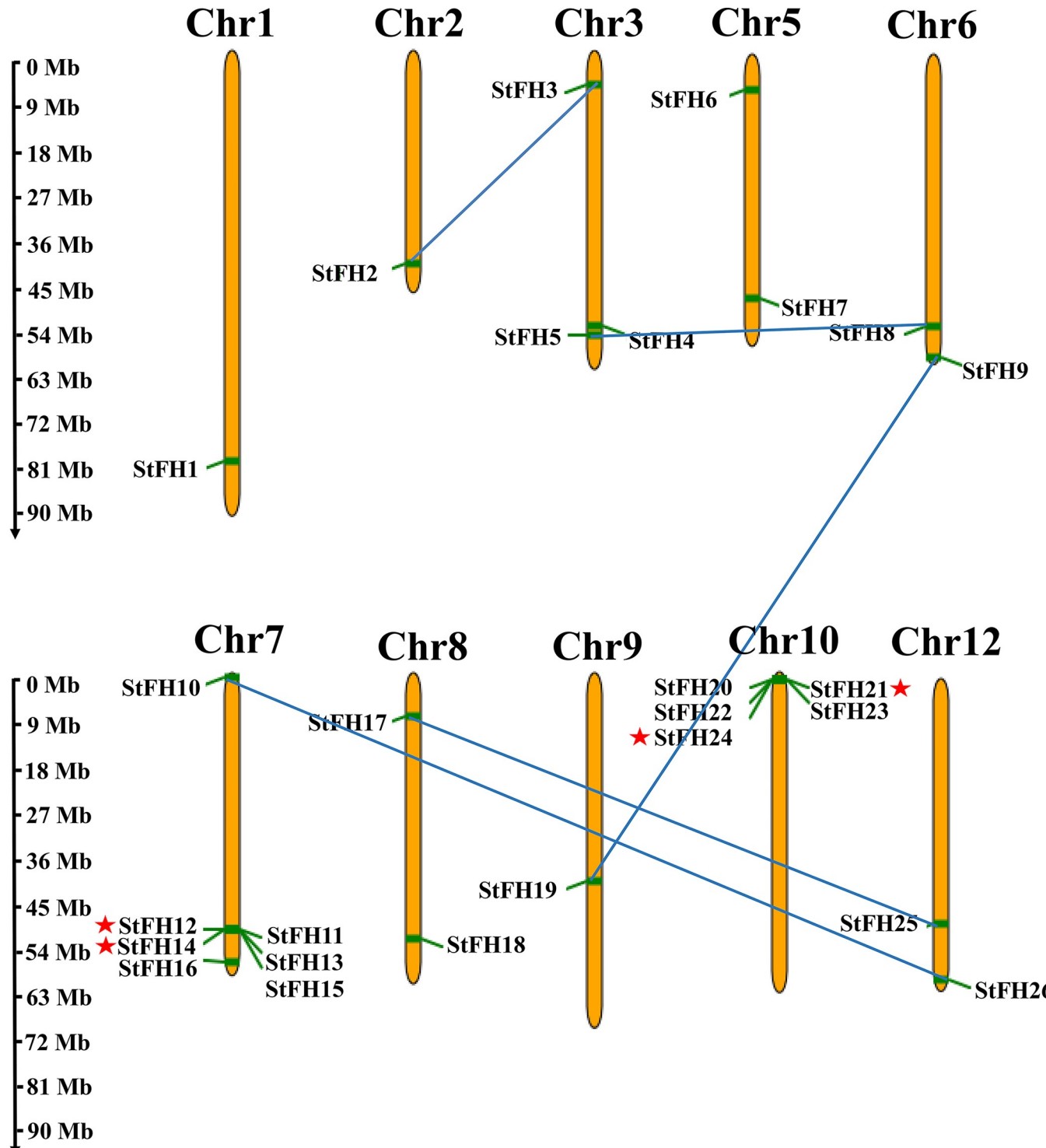

**Fig 8. The chromosomal locations and duplications of *StFH*.** The chromosome-scale is in millions of bases (Mb), showing the length of each chromosome on the left. The chromosomes are colored yellow. The blue lines indicate segmental duplications and the red stars represent tandem duplications.

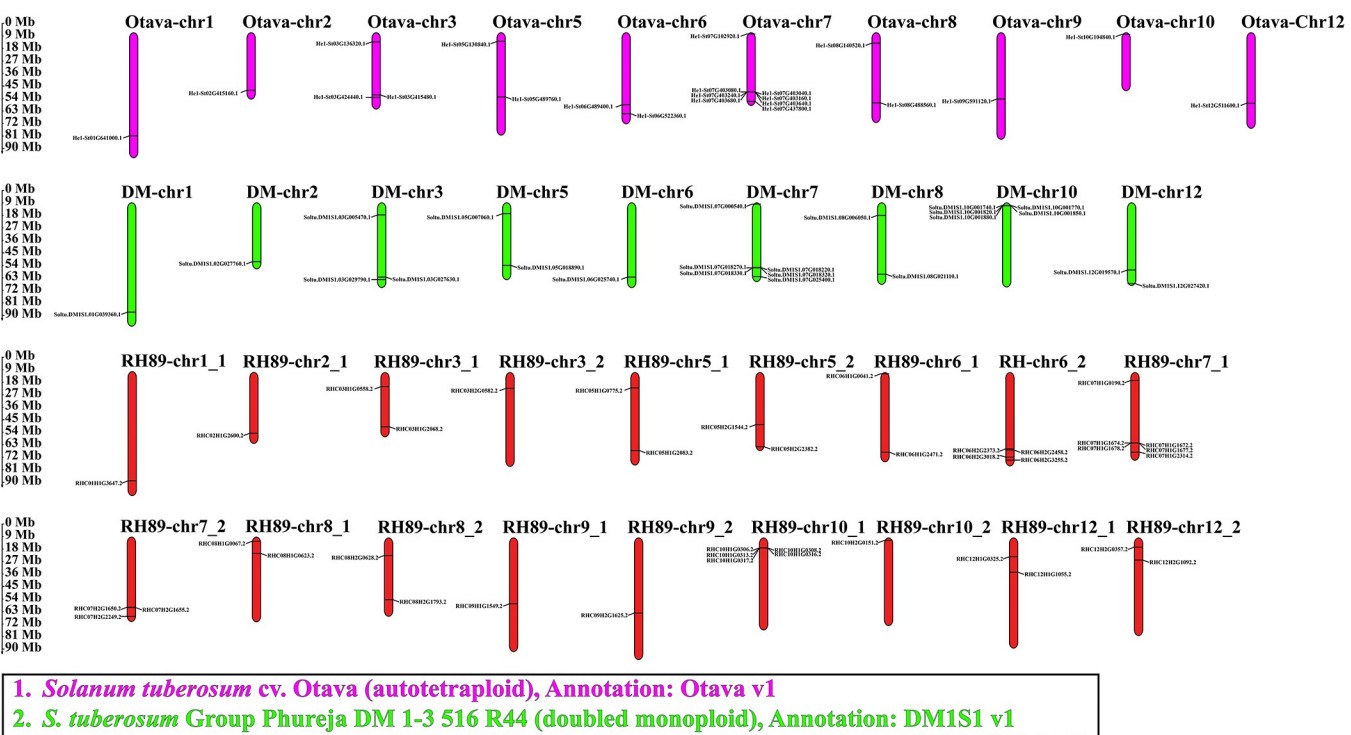

**Fig 9. The distribution of *StFH* genes on multiple sub-genomes of *S. tuberosum*.** The Otava, Phureja DM 1–3516 R44, and Tuberosum RH89-039-16 chromosomes are purple, green, and red, respectively. The chromosome-scale is in millions of bases (Mb).

### 3.12 Gene ontology (GO) analysis of *StFH*

74 GO annotations for *StFH* genes, were classified into three categories: biological process, cellular component, and molecular function. Biological processes were the most predominant, with 63 GO terms identified, including a range of *p*-values from 2.4E-11 to 0.00561 (Fig 12 and S10 Data). The cellular component category encompassed six GO functions with *p*-values ranging from 5.50E-6 to 0.00775. The molecular function category included five GO functions with *p*-values between 4.40E-8 and 0.00029. The most abundant GO annotation in biological processes was "actin filament organization, polymerization, and regulation". In the context of cellular functions, the "membrane part" (GO:0044425; *p*-value: 0.00731) was strongly associated with *StFH* genes. The least number of GO annotations were linked to cellular function, specifically involving "macromolecular complex binding" (GO:0044877; *p*-value: 0.00029).

### 3.13 Transcription factors (TFs) analysis of *StFH*

Thirty-six unique TFs regulating the *StFH* were identified and categorized into seven families: bZIP, C2H2, ERF, GATA, LBD, NAC, and HSF (Fig 13). The ERF family was the largest, comprising 26 TFs (PGSC0003DMG401023951, PGSC0003DMG400018352, PGSC0003DMG400000910, PGSC0003DMG400013402, PGSC0003DMG400009142, PGSC0003DMG400030228, PGSC0003DMG400017813, PGSC0003DMG400002350, PGSC0003DMG400026232, PGSC0003DMG400022305, PGSC0003DMG400014541, PGSC0003DMG400014417, PGSC0003DMG400010719, PGSC0003DMG400003706, PGSC0003DMG400022667, PGSC0003DMG400026136, PGSC0003DMG400007951, PGSC0003DMG400019693, PGSC0003DMG400002185, PGSC0003DMG400028700,

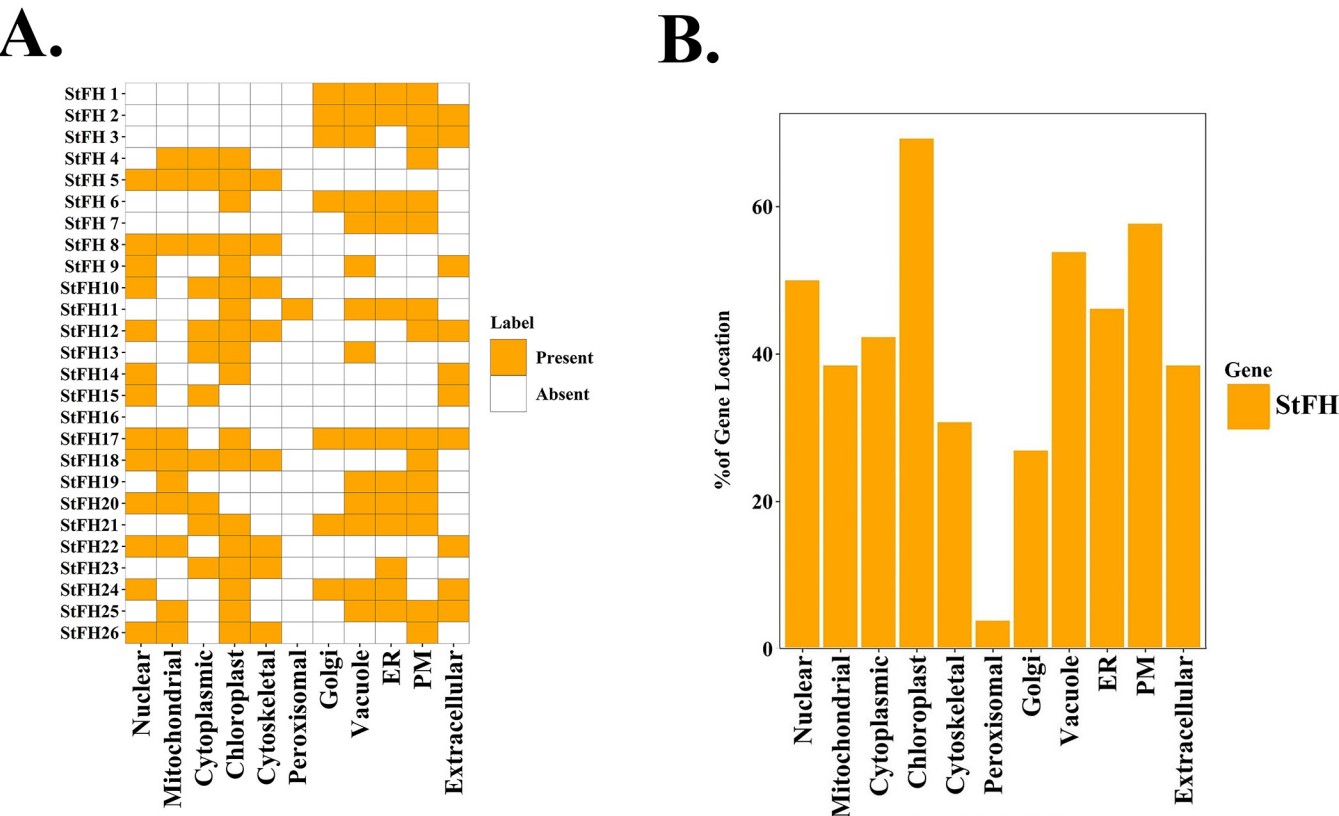

**Fig 10. Sub-cellular localization analysis of StFH.** (A) The heat map represents the sub-cellular localization analysis of StFH proteins. The names of each StFH protein are shown on the left side of the heat map, while the names of the respective cellular organelles are shown at the bottom. The intensity of the color indicates the presence of protein signals corresponding to the genes. The cellular organelles include nuclear, mitochondrial, cytoplasmic, chloroplast, cytoskeletal, peroxisomal, Golgi, vacuole, endoplasmic reticulum (ER), plasma membrane (PM), and extracellular locations. (B) The percentage distribution of StFH protein signals across various cellular organelles is represented by a bar diagram. The percentages of protein signals in different cellular organelles are shown on the left side of the diagram.

PGSC0003DMG400022823, PGSC0003DMG400008352, PGSC0003DMG400040046). The bZIP family was the second largest. with four TFs (PGSC0003DMG400007301, PGSC0003DMG400003529, PGSC0003DMG400003701, PGSC0003DMG400000088). The LBD and NAC families each included three TFs (PGSC0003DMG400018112, PGSC0003DMG400000974, PGSC0003DMG402012772) and (PGSC0003DMG400012113, PGSC0003DMG400009665, PGSC0003DMG400031071), respectively, while the C2H2 and GATA families had one TF (PGSC0003DMG400007202 and PGSC0003DMG400027167, respectively).

### 3.14 Regulatory network between TF and *StFH*

All 36 TF members interact with both the 26 *StFH* genes and each other, revealing a complex network of interactions (Fig 14). The ERF family was prominent, interacting with 20 *StFH* genes, excluding *StFH12*, *StFH13*, *StFH14*, *StFH15*, *StFH22*, and *StFH24*. The bZIP family interacted only with the *StFH25* gene. The C2H2 family interacted with 17 *StFH* genes, excluding several others. The HSF, NAC, LBD, and GATA families interacted with one (*StFH15*), four (*StFH12*, *StFH13*, *StFH14*, and *StFH22*), five (*StFH5*, *StFH6*, *StFH18*, *StFH19*, and *StFH26*), and fourteen *StFH* genes (*StFH1*, *StFH3*, *StFH4*, *StFH5*, *StFH6*, *StFH8*, *StFH9*, *StFH10*, *StFH11*, *StFH17*, *StFH18*, *StFH19*, *StFH25*, and *StFH26*), respectively.

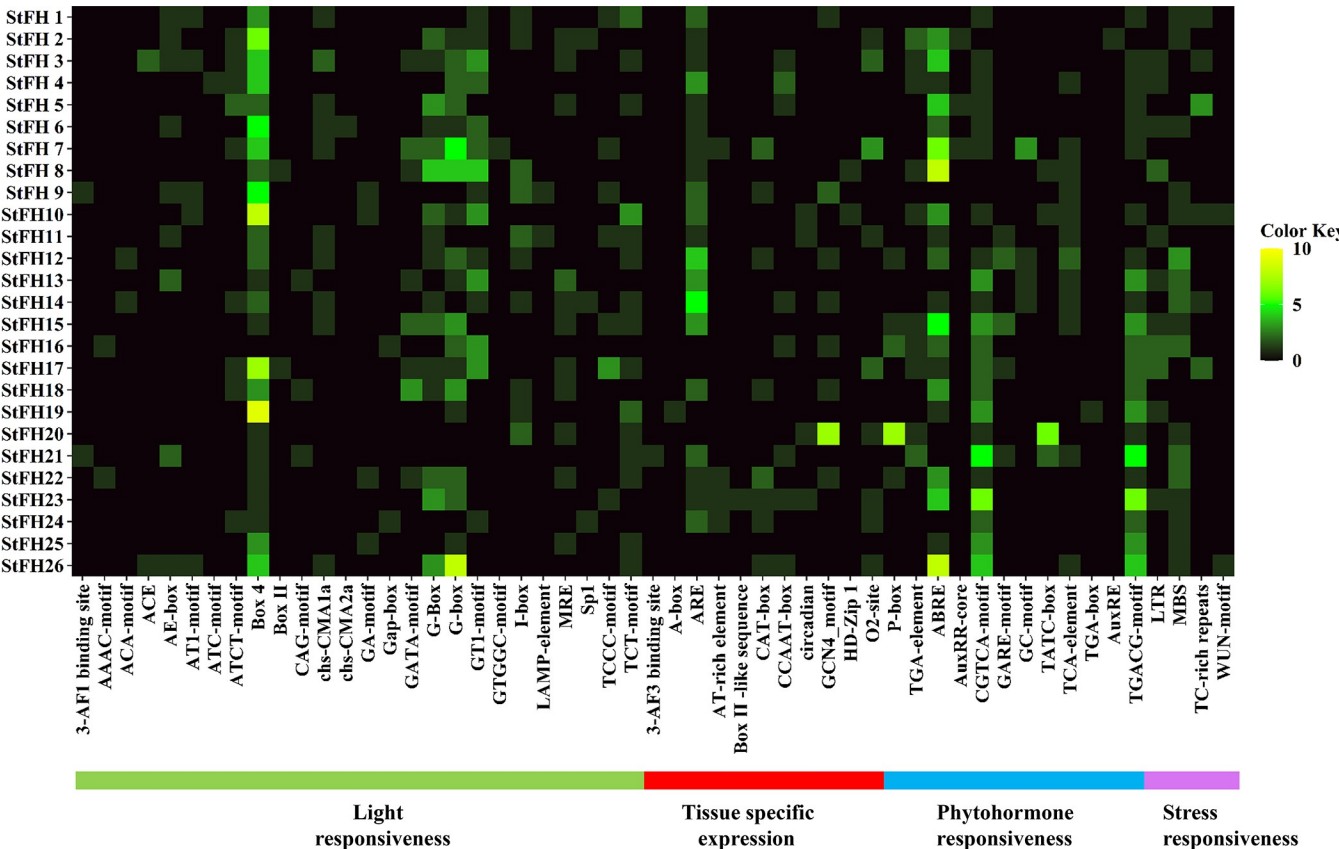

**Fig 11. A heatmap represents the distribution of putative CAREs on the 2.0 kb promoter region of *StFH*.** The names of the CAREs of each *StFH* gene are shown on the left side of the heat map. The number of putative CAREs for each *StFH* gene is displayed on the right side of the heat map and represented by different colors (black = 0, green = 1–5, yellow = 6–10). Functions associated with CAREs of the corresponding genes, such as light responsiveness, tissue-specific expression, phytohormone responsiveness, and stress responsiveness, are shown at the bottom of the heatmap and labeled green, red, blue, and magenta, respectively.

### 3.15 Prediction of potential miRNAs targeting *StFH*

In this analysis, 170 putative miRNAs targeting 24 *StFH* genes were identified. The analysis revealed 60 unique miRNA sequences, with stu-miR156, stu-miR162, stu-miR167, and stu-miR172 being the most abundant (Fig 15A and 15B and S11 Data). These miRNAs targeted various *StFH* genes, with *StFH8*, *StFH10*, *StFH17*, *StFH18*, and *StFH26* being the most frequently targeted.

### 3.16 Protein-protein interactions (PPI) of StFH

The PPI analysis uncovered a strong connection between the 26 StFH proteins and those from *Arabidopsis*, revealing meaningful associations in a biological context (Fig 16 and S12 Data). Specifically, among the StFH proteins, StFH11, StFH14, StFH15, StFH13, StFH16, StFH7, StFH24, StFH22, StFH23, StFH20, and StFH21 exhibited homology with AtFH5, which interacted with FH20, PRF1, PRF2, PRF3, PRF4, and PRF5. Notably, AtFH1 displayed homology with four StFH proteins, namely StFH12, StFH17, StFH25, and StFH9, and interacted with FH14, PRF1, PRF2, PRF3, PRF4, PRF5, and FH13. Furthermore, StFH10 and StFH26 demonstrated homology with AtFH13, which interacted with FH1, PRF2, PRF1, PRF4, and FH5. Meanwhile, AtFH6 exhibited homology with StFH2 and StFH3, and interacted with PRF2,

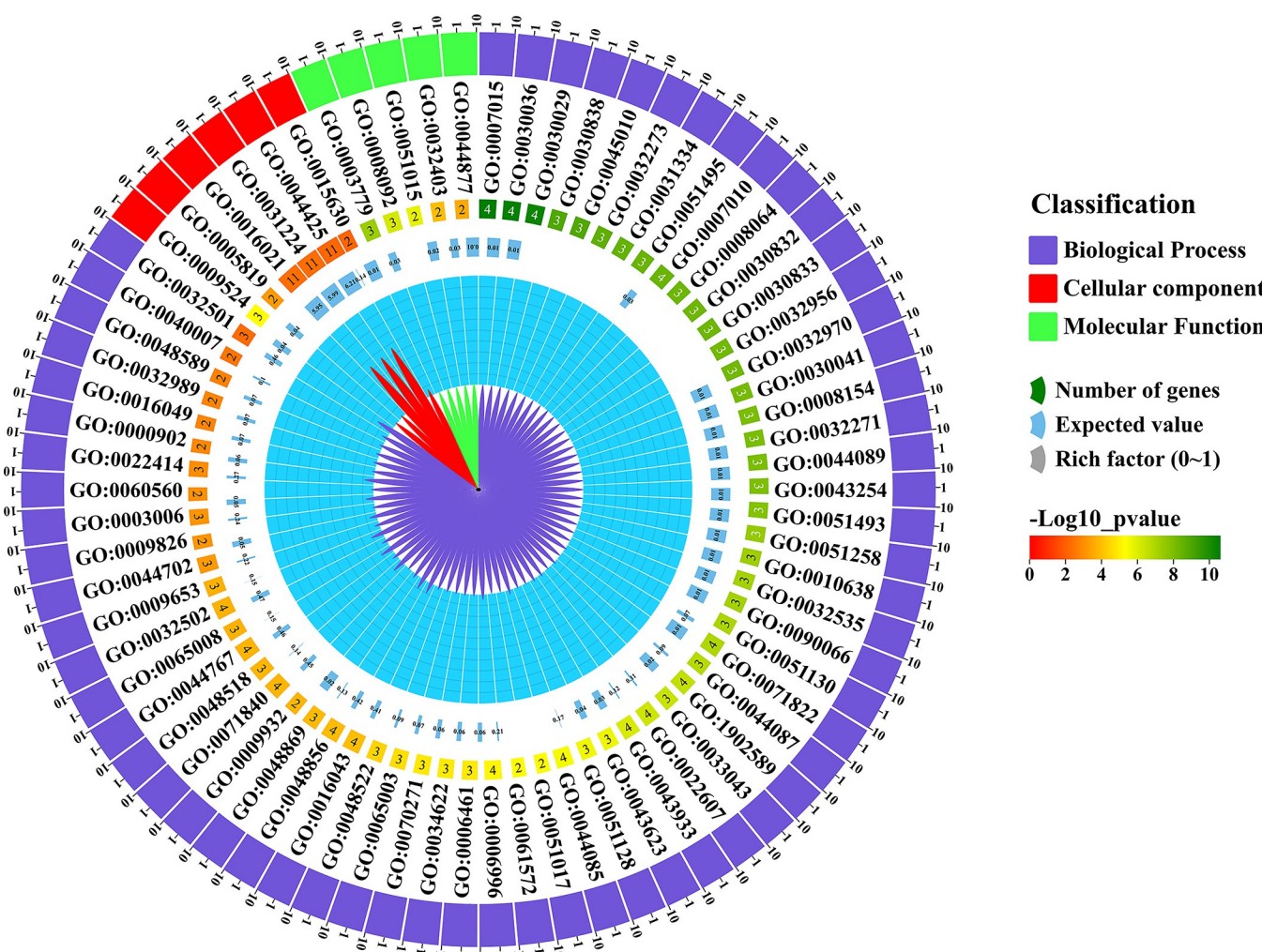

**Fig 12. *StFH* gene function analysis through gene ontology.** The classification of the *StFH* gene function is shown on the right side of the circos plot. The number of genes involved under a specific GO ID, expected value, and rich factor are shown in distinctive colors. The scaling of the -log10 *p*-value is shown in three distinctive colors (red, yellow, and green).

PRF3, PRF4, and PRF5. Additionally, AtFH20 showed homology with StFH5 and StFH8, and AtFH4, AtFH8, AtFH11, and AtFH14 were found to be homologous with StFH6, StFH4, StFH1, and StFH18, respectively. The width or boldness, of the connecting lines between proteins, serves as an indicator of the strength or frequency of their interactions. A broader line connecting two proteins signified a higher interaction ratio, suggesting a more robust and frequent association between the respective proteins. This observation suggested that these StFH proteins played similar roles in biological processes through their interactions with *Arabidopsis* proteins, highlighting potential functional similarities or involvement in cellular activities.

### 3.17 Tissue-specific expression of *StFH*

The tissue-specific expression analysis revealed similar expression levels in floral tissues. In the stamen, *StFH7*, *StFH16*, and *StFH18* were highly expressed compared to other *StFH* genes (Fig 17). Additionally, *StFH2*, *StFH7*, *StFH16*, *StFH18*, and *StFH19* showed higher expression in flowers (S13 Data). The expression levels in leaf tissues were relatively consistent, with *StFH2*,

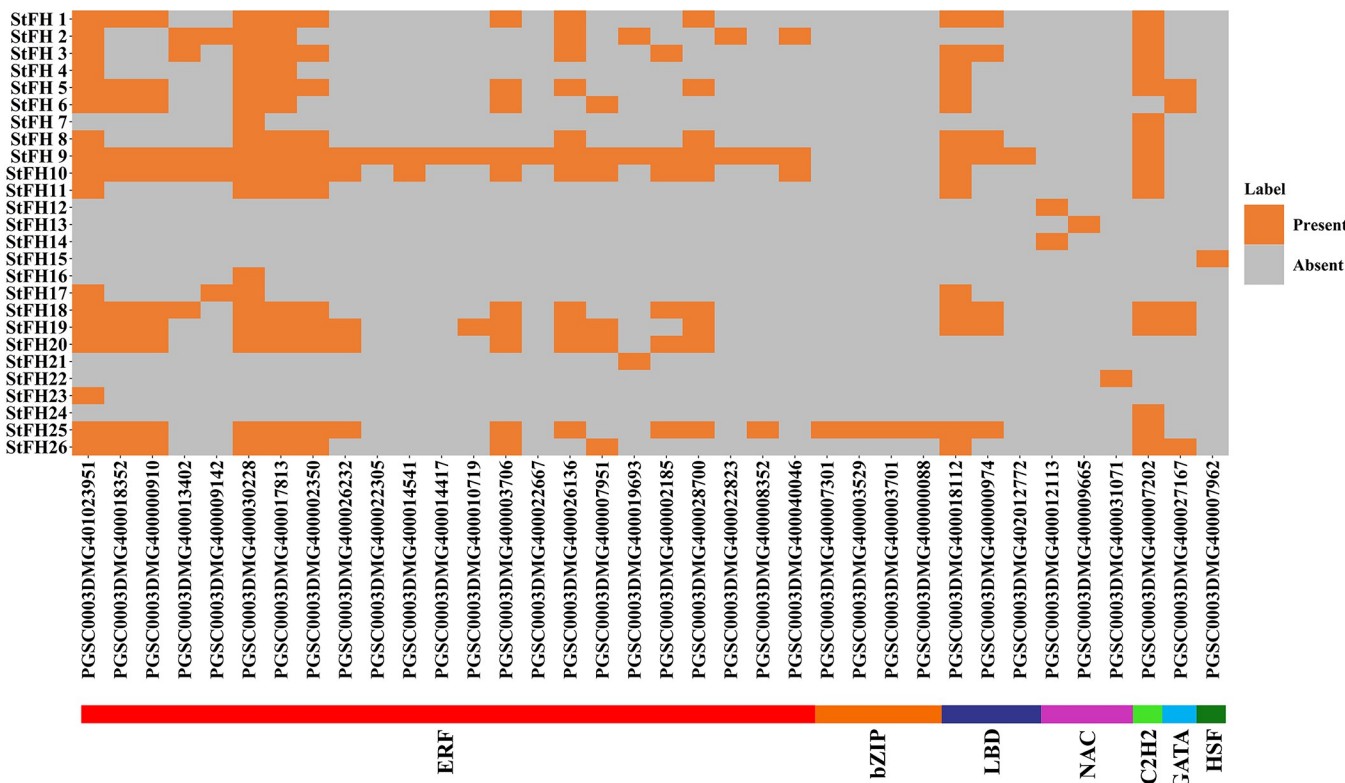

**Fig 13. A heatmap illustrates the TFs regulating *StFH*.** The StFH proteins are on the left, and TF names are at the bottom of the heat map. The color intensity on the right side of the heat map indicates the presence of TFs corresponding to the proteins. Distinctive colors represent the seven TF families: ERF (red), bZIP (orange), LBD (blueberry), NAC (magenta), C2H2 (light green), GATA (sky blue), and HSF (dark green).

*StFH17*, *StFH18*, and *StFH19* being actively expressed. In petiole tissue, *StFH2*, *StFH18*, *StFH19*, and *StFH26* exhibited higher expression levels. The expression levels of *StFH* genes in tuber tissues were similar, with the highest expression observed in the stolon and tuber cortex (88%). Specifically, *StFH1*, *StFH9*, *StFH18*, *StFH19*, and *StFH25* were highly expressed in the stolon. Moreover, *StFH2*, *StFH9*, *StFH18*, *StFH19*, *StFH25*, and *StFH26* displayed similar expression patterns in six tuber tissues (tuber sprout, peel, pith, cortex, young, and mature tuber). *StFH* expression in other tissues such as stem, shoot apex, and root was also similar, with *StFH2*, *StFH3*, *StFH9*, *StFH10*, *StFH18*, *StFH19*, and *StFH25* showing high levels of expression.

### 3.18 Expression pattern of *StFH* under drought stress

The expression patterns of *StFH* genes were not similar in the drought-sensitive Atlantic variety and the drought-resistant Qingshu No. 9 variety. However, only nine *StFH* genes-*StFH1*, *StFH2*, *StFH5*, *StFH10*, *StFH17*, *StFH19*, *StFH21*, *StFH25*, and *StFH26* were up-regulated, while most were down-regulated in Qingshu No. 9 at 25 days after early flowering (Fig 18). At 50 days after full flower blooming, *StFH2*, *StFH5*, *StFH10*, *StFH17*, *StFH19*, *StFH21*, and *StFH25* were up-regulated (S14 Data). However, nine *StFH* genes showed decreased expression levels at 50 days in Qingshu No. 9, indicating down-regulation at the full blooming stage. Moreover, *StFH2*, *StFH5*, *StFH10*, *StFH17*, *StFH19*, *StFH21*, *StFH25*, and *StFH26* were up-regulated in Qingshu No. 9 at 75 days after flower falling. In contrast, *StFH9*, *StFH18*, and *StFH20*

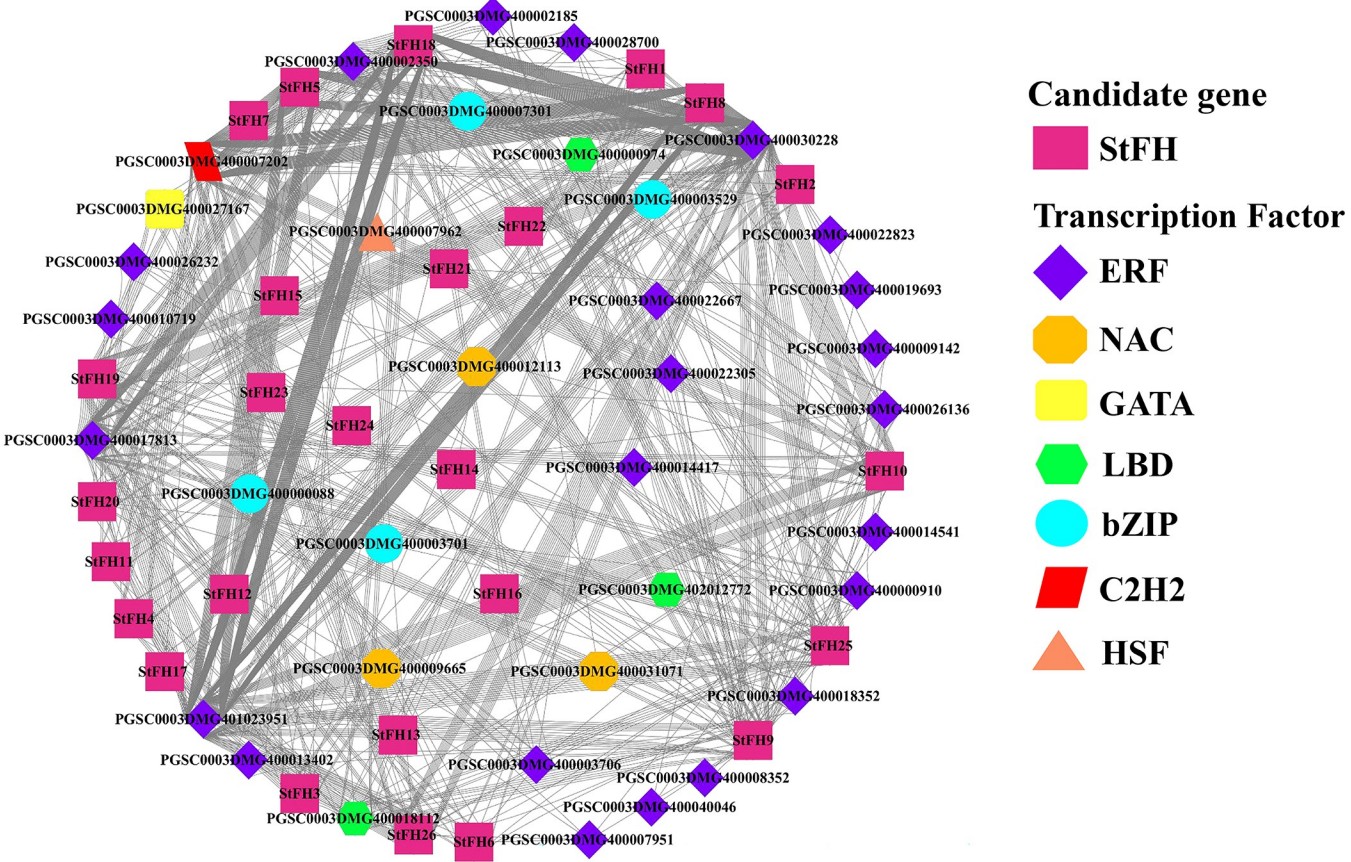

**Fig 14. The regulatory network between TF and *StFH*.** Different colors and shapes represent TFs and their interactions with *StFH* genes. The StFH genes are shown in magenta rectangles, and the TF families are represented by various shapes and colors: ERF (purple diamond), NAC (dark yellow hexagon), GATA (light yellow round rectangle), LBD (lime green hexagon), bZIP (bright aqua ellipse), C2H2 (red parallelogram), and HSF (pastel orange triangle).

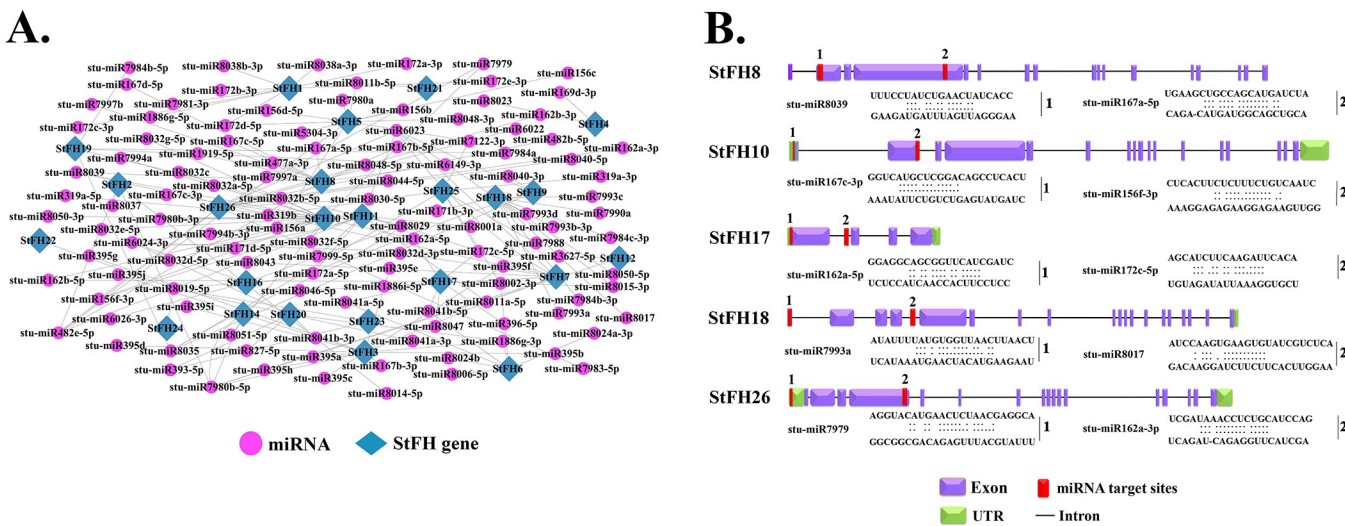

**Fig 15. Prediction of potential miRNAs targeting *StFH*.** (A) The network illustration of predicted miRNA targets shows *StFH* genes in blue-green and miRNAs in light pink ellipses. (B) The schematic diagram indicates the *StFH* genes targeted by miRNAs, with exons (purple), UTR (light green), introns (black lines), and miRNAs (red rectangles).

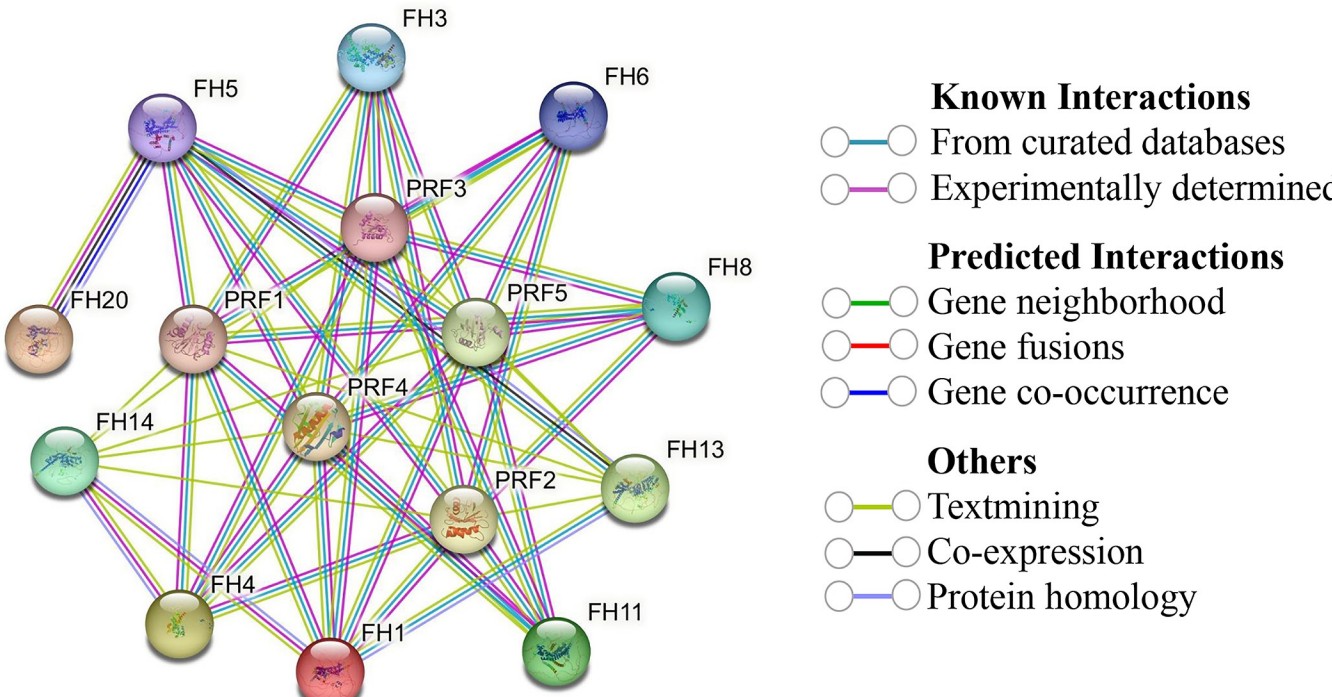

**Fig 16. Protein-protein interaction of StFH based on known *Arabidopsis* proteins.** The network nodes represent proteins, and the line colors indicate different data sources.

were down-regulated in Qingshu No. 9 but exhibited high levels of expression at all three stages in the Atlantic variety.

## 4.0 Discussion

The formin family of proteins is widely distributed in plants and is essential to cellular growth and development [53]. The *de novo* production of actin filaments is regulated through actin cytoskeleton remodeling [54]. Recently, a consistent evolutionary framework has been performed within the Solanaceae family through phylogenetic analyses, which clearly define groups and relationships [55]. Genomic comparisons have revealed a significant expansion of the formin gene family in Solanaceae, particularly within the *Solanum* genus [54]. This expansion is driven by gene duplication events, resulting in a greater diversity of formin proteins with specialized functions. The physiochemical properties of StFH proteins exhibited significant variability in both size and mass, with their chemical nature indicating they were half acidic and half alkaline. Moreover, 22 StFH proteins are potentially unstable, as their instability index surpassed 40.0, suggesting their susceptibility to structural fluctuations [56]. Moreover, the aliphatic index indicated extensive diversity among StFH proteins. The negative GRAVY scores indicate a hydrophilic disposition and a preference for aqueous environments [57, 58]. Therefore, all the StFH proteins were hydrophilic.

The phylogenetic relationships play a crucial role in uncovering the molecular basis and evolutionary aspects of lineages, interactions, and diversity within various species [59]. This analysis revealed the patterns and rates of evolution for diverse species. The findings revealed significant clustering patterns, demonstrating that a substantial portion of StFH proteins formed clusters closely associated with AtFH and MtFH. The gene organization demonstrated

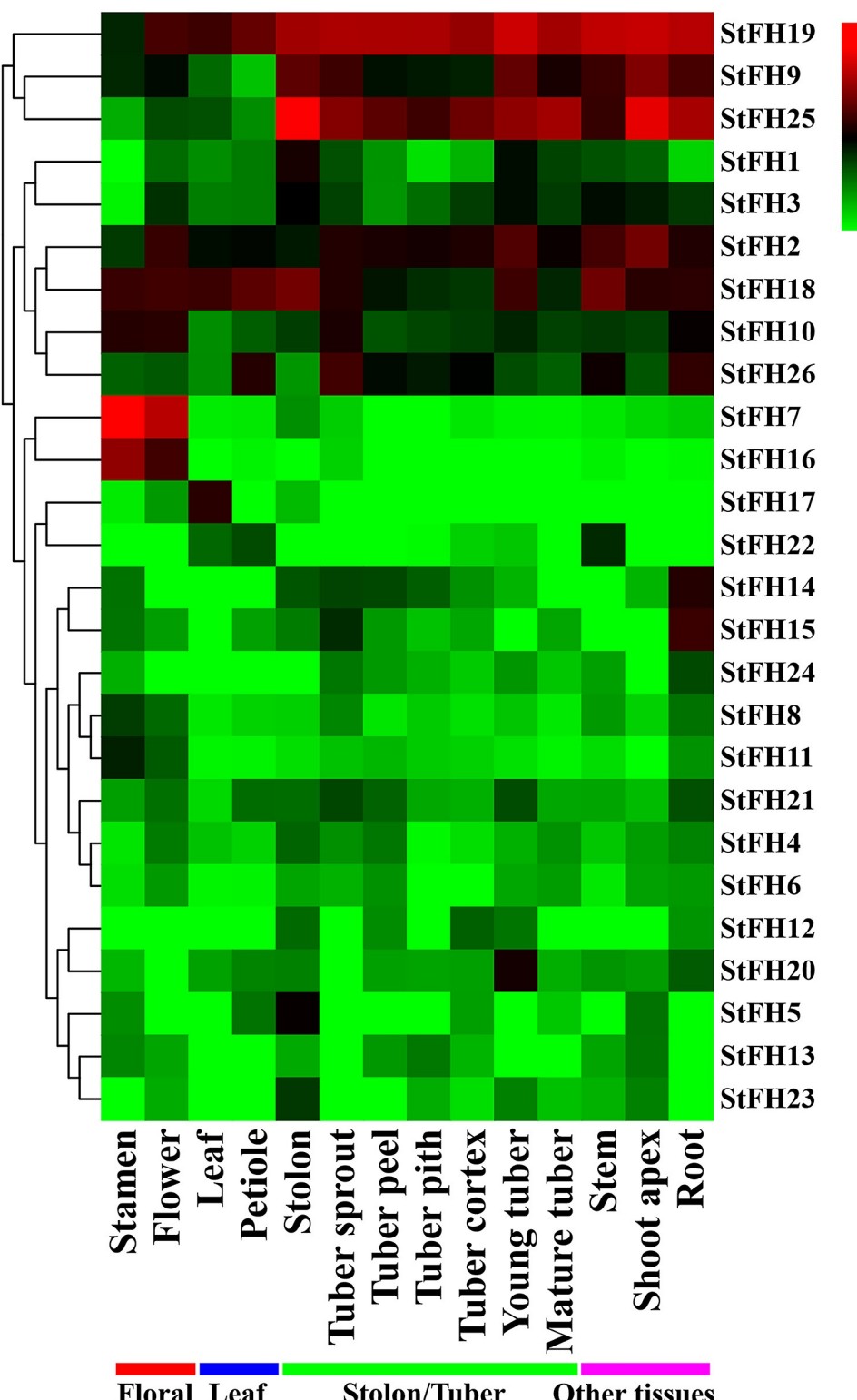

**Fig 17. Tissue-specific expression pattern of *StFH*.** The respective *StFH* gene names are shown on the right side of the heat map. Various tissues are represented at the bottom of the heat map. The color gradient from green to red indicates the expression levels on the right side of the heat map.

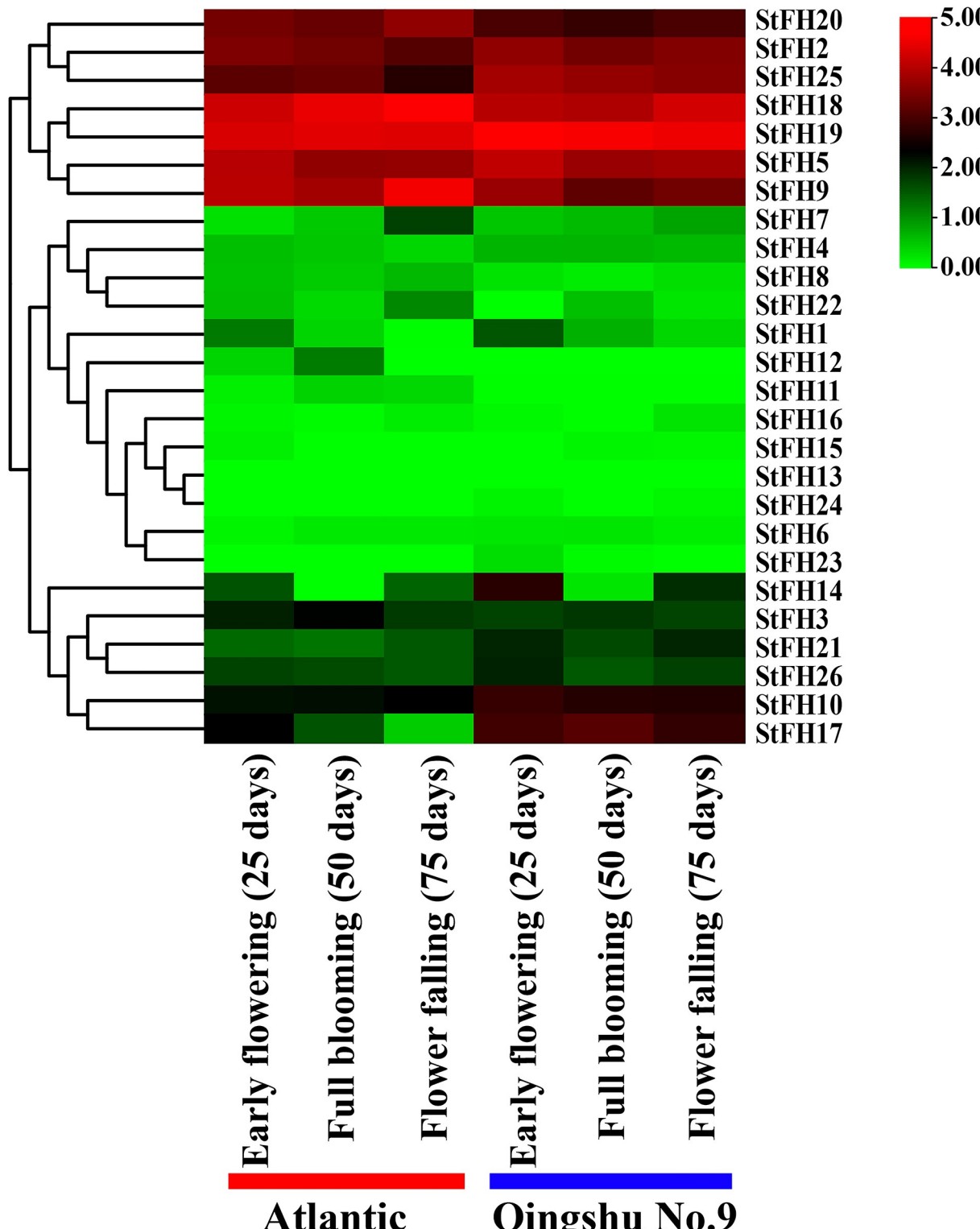

**Fig 18. Expression pattern of *StFH* under drought stress.** The respective *StFH* gene names are shown on the right side of the heatmap. The stages of the plant in two varieties (Atlantic and Qingshu No. 9) are represented at the bottom of the heat map. The color gradient from white to red indicates the expression levels on the right side of the heat map.

that many introns allow for alternative splicing, which can be linked to specific and significant biological roles [60, 61]. Thus, *StFH26* was involved in alternative splicing. Besides, genes with higher expression levels tend to possess longer introns, distinguishing them from low-expression genes, and reduced exon numbers were often marked as early-responsive genes [62, 63]. Therefore, *StFH4*, *StFH6*, and *StFH13* genes exhibited reduced exon numbers, marking them as early response genes, aligning with their higher activation potential.

The FH2 domain predominantly plays a crucial role in cellular processes, particularly in the regulation of actin dynamics [64, 65]. Thus, the presence of the FH2 domain in all the StFH proteins suggested that they might have a role in cytoskeleton regulation. A conserved motif is a repeated and evolutionary preserved sequence pattern in proteins with vital biological functions [66]. The identified motifs in the *StFH* showed differences between groups while being relatively similar within the same group. For instance, the eleven motifs of *StFH9* and *StFH19* in group G were quite similar, suggesting their potential involvement in relevant biological processes. In contrast, other groups exhibited variations in these motif sets [67]. The Ka/Ks ratios among *StFH* gene pairs were less than 1, suggesting a purifying selection process within the gene family. Moreover, the evolutionary time divergence of *StFH* gene pairs revealed that they emerged between 18.96 to 86.51 MYA (million years ago) [68].

Chromosomal localization provides mapping of genes to specific locations on chromosomes. Moreover, *OsFH* genes were distributed on nine chromosomes [69]. Consequently, twenty-six *StFH* genes with the highest number of genes observed on chromosome 7, were mapped to 10 chromosomes. Besides, the presence of the highest number of genes on chromosome 7 in the three sub-genomes of *S. tuberosum* also highlighted its similarity. Thus, it sheds light on genome evolution, gene function, phenotypic variation, and improvement [70]. Besides, autotetraploid plants have shown higher drought tolerance and heterozygosity compared to diploids [71, 72]. The gene duplication mechanisms create genetic redundancy, allowing one copy of a gene to evolve new functions while the other maintains its original function. Tandem, whole genome, and segmental duplications are the main mechanisms for expanding the gene families in many plant species [73, 74]. The segmental and tandem duplications were pivotal in the expansion of the *StZFP* gene family [75]. Thus, five segmental and two tandem duplication events observed in *StFH* suggest that they persisted over time, most likely due to their significant functional importance. Furthermore, the syntenic comparison of *S. tuberosum* with three species (two monocotyledons and one dicotyledon) demonstrated a close evolutionary relationship and a higher degree of resemblance in genomic conservation with the dicotyledon, *A. thaliana*.

Formin proteins are critical actin filament builders and display various subcellular localization patterns based on their particular isoforms and cellular activities [76]. Additionally, some formins, like Fmn1 and mDia1, interact with microtubules and may coordinate cytoskeleton dynamics as well as regulate gene expression [77, 78]. The subcellular localization analysis revealed that StFH protein signals were most abundant in the chloroplast and plasma membrane, followed by the vacuole and nucleus.

Generally, GO is analyzed to distinguish the functions of individual genes into three categories; biological processes, cellular components, and molecular functions [79, 80]. This study identified most of the *StFH* genes engaged in biological processes regarding actin filament polymerization. CAREs are DNA sequences found in gene promoter regions, controlling gene expression and binding TFs [81]. They play crucial roles in biological processes like development, adaptation, and response to environmental stresses [82]. Most of the CARE motif in *StFH* was related to light response. However, some stress-responsive elements were also found. The stress responsiveness elements included LTR, MBS, TC-rich repeats, and WUN-motif, responsible for managing abiotic stress such as drought [83]. In *Arabidopsis*, *AtMYB2* and

*AtMYB60* were involved in drought stress [84, 85]. The TFs play crucial roles in various biological functions, specifically regulating metabolism, promoting growth, facilitating progression, providing resistance against microbial infections, and responding to both biotic and abiotic stress [86, 87]. Among the seven TF families identified in *StFH*, the ERF (Ethylene Response Factor) TF in potatoes explores their role in negatively regulating specific processes [88]. Whereas, bZIP (basic Leucine Zipper) TF might confer resistance against abiotic stress such as dehydration, drought, and salt [67]. Moreover, drought-responsive elements, for instance, NAC TFs, are primarily identified as binding to drought-related transposons in citrus [89]. Furthermore, the up-regulation of HSF, GTE, DREB2B, bHLH, MYB, HD-ZIP, and ERF TF families during heat and cold stress in potatoes was observed [90]. Besides, the regulatory network between TFs and *StFH* interacts with each other, revealing a complex network of interactions.

The pivotal role of PPIs in shaping the evolution and functionality of living organisms has been well established [91]. It provides insights into species diversification and the complex regulation of cellular functions, which allow for the coordination and communication of diverse cellular processes [92]. The StFH showed homology with *Arabidopsis* and interacted with the FH family (FH1, FH3, FH4, FH5, FH6, FH8, FH11, FH13, FH14, and FH20) and PRF (Profilin) family (PRF1, PRF2, PRF3, PRF4, and PRF5). The FH family proteins are primarily recognized as cytoskeletal dynamics regulators, emerging as potential actin nucleation agents. Whereas PRF, a low-molecular-weight actin-binding protein, plays a crucial role in plant development during cell elongation and division [93–95]. The role of miRNA is not confined to cellular signaling, as it has potential involvement in various abiotic stresses such as heat, salinity, low temperature, drought, and biotic stresses like viral and bacterial attack [96]. Further, 60 unique miRNAs were identified to control essential biological processes in *StFH*, particularly their potential role in abiotic stress management. The crucial functions of stu-miR156 in the biological processes of potatoes, such as tuberization, can lead to the formation of aerial tubers under specific conditions (Table 2). Moreover, miR156 plays a vital role in controlling different aspects, notably influencing the development of lateral roots in potatoes [97]. Besides, miR156 is induced by drought stress [98]. Besides, Stu-miR172, a phloem-mobile miRNA, plays a crucial role in sugar-dependent signal transduction pathways, influencing flower and tuber induction [99]. However, the most pivotal role of miR172 in potatoes is managing drought stress [100]. Meanwhile, stu-miR162 in potatoes regulates miRNA biogenesis, plant development, and abiotic stress responses [101]. Stu-miR167 in potatoes is involved in

**Table 2. Information about abundant miRNA ID, functions, and their targeted *StFH* genes.**

| miRNA ID | Functions | Targeted genes |
|---|---|---|
| stu-miR156 | It influences the development of lateral roots in potatoes as well as links to the regulatory mechanisms involved in tuberization. It is enhanced by drought stress. | *StFH5, StFH10, StFH11, StFH26* |
| stu-miR162 | miRNA biogenesis, plant development, and abiotic stress responses. | *StFH10, StFH17, StFH26* |
| stu-miR167 | Its positively regulates nuclear factor Y subunit A (NF-YA) and flavin-binding monooxygenase family protein (YUC2), activating the auxin signaling pathway. | *StFH3, StFH8, StFH10* |
| stu-miR172 | It is a sugar-dependent signal transduction pathway and influences flower and tuber induction. It also regulates the transitions between developmental stages and specifies floral organ identity. It activates under drought conditions. | *StFH1, StFH5, StFH11, StFH17, StFH25* |

positively regulating nuclear factor Y subunit A (NF-YA) and flavin-binding monooxygenase family protein (YUC2), activating the auxin signaling pathway [102].

Tissue-specific gene expression is a core biological mechanism that allows plants to respond more efficiently to stress by stress-responsive genes in specific cells or tissues, such as roots or leaves [103, 104]. For example, certain tissues like the root (which functions as a water and nutrient uptake system and can produce extensive lateral roots and deep root hair), the stem (which transports water, nutrients, and hormones), and the shoot apex (a center for auxin biosynthesis and growth regulation) are involved in sensing and initiating signaling cascades to counteract drought [105–108]. Similarly, the expression of *StRFP2* in drought-responsive tissues (root, stem, and shoot apex) was observed [109]. Therefore, the expression of *StFH2*, *StFH3*, *StFH9*, *StFH10*, *StFH18*, *StFH19*, *StFH25*, and *StFH26* in these drought-responsive tissues indicates a key role in drought tolerance. Moreover, leaf tissues contain stomata (which are centers for $CO_2$ and $O_2$ gas exchange) and guard cells (which regulate the opening and closing of stomata) to counteract drought stress [110]. In potatoes, the expression of *StPIP1* in leaves was found to be involved in managing drought stress [111]. Consequently, the expression of *StFH1*, *StFH18*, and *StFH19* in leaf tissue suggests their importance in enhancing drought tolerance. Other tissues, such as the stolon (responsible for forming tubers) and tuber (storehouse of carbohydrates), also play potential roles in plant growth, development, and metabolism under drought stress [112, 113]. The various expressions of *StFH* genes in these tissues highlighted their significance. The drought-resistant Qingshu No. 9, a high-yielding variety, further supports the significance of gene expression patterns in response to drought [114, 115]. The up-regulation of *StFH2*, *StFH5*, *StFH10*, *StFH17*, *StFH19*, *StFH21*, and *StFH25* underscores their important roles in managing drought stress. Specifically, *StFH2*, *StFH10*, *StFH19*, and *StFH25* exhibited significant expression in various tissues when responding to drought stress through up-regulation. In summary, this study emphasizes the crucial role of *StFH* genes in conferring drought tolerance and developing new strategies for breeding programs aimed at developing drought-resistant potato varieties.

## 5.0 Conclusion

In this study, 26 *StFH* genes were identified in the potato genome, distributed in ten chromosomes. These genes show ancestry and functional resemblance to the dicotyledon *Arabidopsis*, revealing similarities in their gene structures, typical domains, and motifs. The observation of both segmental and tandem duplications indicates the expansion of this gene family. The divergence time and evolutionary relationship analysis suggest that *StFH* genes have evolved through purifying selection, maintaining functional stability. Moreover, CARE analysis revealed the binding of major TFs such as ERF, bZIP, and C2H2 in the promoter region of *StFH*, indicating their role in regulating gene expression under various abiotic stresses, particularly drought. Most of the *StFH* genes were found to perform biological functions, and their expression was regulated by miRNAs under abiotic stress conditions such as drought. The expression of *StFH2*, *StFH3*, *StFH9*, *StFH10*, *StFH18*, *StFH19*, *StFH25*, and *StFH26* in major drought-responsive tissues such as root, stem, and shoot apex indicated their involvement in stress response. The RNA-seq data confirmed the potential role of *StFH2*, *StFH10*, *StFH19*, and *StFH25*, as these genes showed significant up-regulation in the drought-resistant variety. Overall, the findings from this study provide valuable insights into candidate gene selection, gene function validation, stress mechanisms, and the development of stress-tolerant potato cultivars for future breeding programs.

## Supporting information

**S1 Data. Full-length protein sequences of *StFH* gene families of *S. tuberosum* plant species.**
(TXT)

**S2 Data. Full-length protein sequences of FH2 gene families of *S. tuberosum*, *M. truncatula*, *Z. mays*, *O. sativa*, *L. japonicus*, and *A. thaliana* plant species for constructing a phylogenetic tree.**
(TXT)

**S3 Data. Full-length coding sequences of *StFH* gene families of *S. tuberosum* plant species.**
(TXT)

**S4 Data. Full-length genomic sequences of *StFH* gene families of *S. tuberosum* plant species.**
(TXT)

**S5 Data. The 5′ untranslated region (5′ UTR) (2.0 sequences) of *StFH* gene families of *S. tuberosum* for the analysis of *cis*-acting regulatory elements (CAREs).**
(TXT)

**S6 Data. StFH protein family distribution among groups based on phylogenetic analysis with *Z. mays*, *O. sativa*, *A. thaliana*, *M. truncatula*, and *L. japonicas* FH2 members.**
(DOCX)

**S7 Data. *In silico* predicted the number of introns and exons in *StFH* genes.**
(DOCX)

**S8 Data. Time of gene duplication estimated for different pairs of *StFH* genes based on Ka and Ks values.**
(XLSX)

**S9 Data. The predicted *cis*-acting regulatory elements (CAREs) of the *StFH* gene.**
(XLSX)

**S10 Data. The GO analysis of *StFH* gene families for the identification of gene functions in *S. tuberosum* plant species.**
(XLSX)

**S11 Data. The putative miRNA identification of *StFH* gene families in *S. tuberosum* plant species.**
(XLSX)

**S12 Data. Protein-protein interactions of *StFH* gene families in *S. tuberosum* plant species.**
(XLSX)

**S13 Data. Expression patterns of *StFH* genes in different tissues in *S. tuberosum* plant species.**
(XLSX)

**S14 Data. Expression patterns of *StFH* genes in drought susceptible variety Atlantic and resistant variety Qingshu No. 9.**
(XLSX)

**S1 Fig. Bubble plot of subcellular localization of *StFH* genes.**
(TIFF)

## Acknowledgments

The opportunity to conduct this research provided by the Laboratory of Functional Genomics and Proteomics, Department of Genetic Engineering and Biotechnology, Faculty of Biological Science and Technology, Jashore University of Science and Technology, Jashore 7408, Bangladesh, is gratefully acknowledged by the authors. Gratitude is extended to Mr. Tanzir Ahmed, Assistant Professor, Department of English, Faculty of Arts and Social Science, Jashore University of Science and Technology, Jashore 7408, Bangladesh for extensively editing the manuscript to avoid grammatical errors. The valuable comments and critical suggestions for improving the quality of this manuscript provided by the reviewers and members of the editorial panel are also acknowledged and appreciated.

## Author Contributions

**Conceptualization:** Md. Abdur Rauf Sarkar.

**Data curation:** Mst. Sumaiya Khatun, Md Shohel Ul Islam, Fatema Tuz Zohra, Shuraya Beente Rashid, Md. Abdur Rauf Sarkar.

**Formal analysis:** Mst. Sumaiya Khatun, Md Shohel Ul Islam, Md. Abdur Rauf Sarkar.

**Methodology:** Mst. Sumaiya Khatun, Md Shohel Ul Islam, Fatema Tuz Zohra, Md. Abdur Rauf Sarkar.

**Supervision:** Md. Abdur Rauf Sarkar.

**Visualization:** Mst. Sumaiya Khatun, Md Shohel Ul Islam, Pollob Shing, Shuraya Beente Rashid, Md. Abdur Rauf Sarkar.

**Writing – original draft:** Mst. Sumaiya Khatun, Md Shohel Ul Islam, Pollob Shing, Fatema Tuz Zohra, Shuraya Beente Rashid, Shaikh Mizanur Rahman, Md. Abdur Rauf Sarkar.

**Writing – review & editing:** Mst. Sumaiya Khatun, Md Shohel Ul Islam, Pollob Shing, Fatema Tuz Zohra, Shuraya Beente Rashid, Shaikh Mizanur Rahman, Md. Abdur Rauf Sarkar.

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
