## [Decision Letter · Decision Letter 0]

7 May 2024

PONE-D-24-14310Genome-wide identification and characterization of FORMIN gene family in potato (Solanum tuberosum L.) and their expression profiles in response to drought stress conditionPLOS ONE

Dear Dr. Sarkar,

Thank you for submitting your manuscript to PLOS ONE. After careful consideration, we feel that it has merit but does not fully meet PLOS ONE’s publication criteria as it currently stands. Therefore, we invite you to submit a revised version of the manuscript that addresses the points raised during the review process.

Please revise the manuscript carefully, point by point, according to the reviewer's comments. Additionally, improve the English language of the manuscript according to journal standards.

We look forward to receiving your revised manuscript.

Kind regards,

Muhammad Anwar, PHD

Academic Editor

PLOS ONE

Journal Requirements:

Reviewers' comments:

Reviewer's Responses to Questions

**Comments to the Author**

1. Is the manuscript technically sound, and do the data support the conclusions?

Reviewer #1: Yes

Reviewer #2: Yes

2. Has the statistical analysis been performed appropriately and rigorously? 

Reviewer #1: Yes

Reviewer #2: Yes

3. Have the authors made all data underlying the findings in their manuscript fully available?

Reviewer #1: Yes

Reviewer #2: Yes

4. Is the manuscript presented in an intelligible fashion and written in standard English?

Reviewer #1: No

Reviewer #2: Yes

5. Review Comments to the Author

Reviewer #1: In this manuscript, authors identified a total of 26 StFH genes in the S. tuberosum genome, and analyzed their expression profiling, subcellular location, duplication and selection pattern, miRNA targeting prediction, and protein-protein interaction. However, there are a few points which should be improved, as delineated below.

[1] The well-crafted professional editing service should be employed for this manuscript. Additionally, it is necessary to reduce the number of words in this Abstract, owing to 300-word limit in the PlosONE’s Abstract.

[2] The authors should provide tissue/organ-specific and drought-responsive expression pattern of StFH genes using RT-qPCR.

[3] To further validate the predicted results, it is necessary to provide an experimental evidence for the subcellular localization of StFHs.

[4] The version of tools, such as BlastP (Line121), MEGA11 (L139), ClustalW (L140), Trimmonmatic (L231), bowtie2 (L234), Samtools (L235) and RSEM (L236), should be replenished in the ‘Materials and Methods’ section.

[5] The splice-aware mappers (e.g., TopHat2/HISAT2, STAR) are capable of mapping RNA-seq reads against the reference genome. However, the Bowtie2 is not splice-aware aligner, and used frequently for the mapping of DNA-seq reads to the reference genome. Bowtie2, therefore, is not suitable for alignment of RNA-seq reads to the S. tuberosum genome (L234).

[6] In Fig.1, the bootstrapping value should be added in the branch of the maximum likelihood (ML) tree. Compared to MEGA, IQ-TREE or RAxML might be a more appropriate tool for reconstructing ML tree, especially in the choice of the best-fitting substitution model.

[7] In Fig.5, StFH20 and StFH22 were identified as a duplicated gene pair. It might be problem, owing to a great difference in their CDS length (Fig.3, > 3 fold). DupGen_finder or MCScanX should be a toolkit for detection of duplicated gene pairs.

[8] The gene name should be italicized, e.g., StFH (Line36, L40, L848, L851, L852, L854).

Reviewer #2: The paper titled: " Genome-wide identification and characterization of FORMIN gene family in potato (Solanum tuberosum L.) and their expression profiles in response to drought stress condition " reported the identification, characterization and of FORMIN gene family in Solanum tuberosum and the expression patterns of FORMIN members using RNA-seq data, offering a foundation for further research aiming at potato enhancement. Overall, the paper is clearly presented. The authors are invited to revise the English and the sentences structure in the manuscript before publication. Also, genes must be written in Italic in the whole text.

6. PLOS authors have the option to publish the peer review history of their article (what does this mean?). If published, this will include your full peer review and any attached files.

Reviewer #1: No

Reviewer #2: **Yes: **Hatem Boubakri

---

## [Author Response · Author response to Decision Letter 0]

1 Jun 2024

Response to Reviewers

Reviewers' comments:

Responses to Reviewer's Questions

Review Comments to the Author

Reviewer #1:

In this manuscript, authors identified a total of 26 StFH genes in the S. tuberosum genome, and analyzed their expression profiling, subcellular location, duplication and selection pattern, miRNA targeting prediction, and protein-protein interaction. However, there are a few points which should be improved, as delineated below.

Comment #1: The well-crafted professional editing service should be employed for this manuscript. Additionally, it is necessary to reduce the number of words in this Abstract, owing to 300-word limit in the PlosONE’s Abstract.

Responses to Comment #1: Thank you for your valuable suggestion to improve our manuscript for language usage, spelling, and grammar by getting English editing and proofreading services. We are pleased to inform you that, to improve our manuscript at a standard level and to avoid grammatical errors, we are thoroughly edited and proofreading of our full manuscript by an academic English professional named Mr. Tanzir Ahmed, Assistant Professor, Department of English, Faculty of Arts and Social Science, Jashore University of Science and Technology, Jashore 7408, Bangladesh (https://just.edu.bd/t/teacher-1549444621686), email: tanzir@just.edu.bd which are highlighted as a track change options (Please see the track changes area around the full manuscript). We also acknowledged his name in the acknowledgement section of our manuscript (Please see the lines 833-836). Hopefully, the language of the revised manuscript is now more standard than the previous one. 

Moreover, we have reduced the word number below 300 limits to PlosONE’s Abstract style.

Comment #2: The authors should provide tissue/organ-specific and drought-responsive expression pattern of StFH genes using RT-qPCR.

Responses to Comment #2: Thank you for your valuable comment and suggestion. We appreciate your insights and fully agree with the importance of validating the candidate StFH genes using RT-qPCR through wet lab experiments. We would like to inform you that, currently we do not have a enough research grant to support the experimental validation of the data in our laboratory or to seek assistance from other well-equipped facilities. We sincerely apologize for the inconvenience caused regarding RT-qPCR experiment and hope for your understanding and consideration regarding this issue.

Comment #3: To further validate the predicted results, it is necessary to provide experimental evidence for the subcellular localization of StFHs.

Responses to Comment #3: Thank you for your valuable necessary comment and suggestion. We also agree with the importance of provide experimental evidence for the subcellular localization of StFHs experimental evidence for the subcellular localization of StFHs through wet lab experiments. We cordially would like to inform you that, currently we do not have a research grant to support the experimental validation of the data in our laboratory or to seek assistance from other well-equipped facilities. We sincerely apologize for the inconvenience caused regarding experimental evidence for the subcellular localization of StFHs and hope for your understanding and consideration regarding this issue.

Comment #4: The version of tools, such as BlastP (Line121), MEGA11 (L139), ClustalW (L140), Trimmonmatic (L231), bowtie2 (L234), Samtools (L235) and RSEM (L236), should be replenished in the ‘Materials and Methods’ section.

Responses to Comment #4: Thank you very much for your kind suggestions. According to your comment, we have replaced the above-mentioned tools explanation in the ‘Materials and Methods’ section of our revised manuscript. Please see the revised section of ‘Materials and Methods’ and lines 119-124, and lines 208-214.

Comment #5: The splice-aware mappers (e.g., TopHat2/HISAT2, STAR) are capable of mapping RNA-seq reads against the reference genome. However, the Bowtie2 is not splice-aware aligner, and used frequently for the mapping of DNA-seq reads to the reference genome. Bowtie2, therefore, is not suitable for alignment of RNA-seq reads to the S. tuberosum genome (L234).

Responses to Comment #5: Thank you for your valuable comment on alignment of RNA-seq reads to the S. tuberosum genome for expression pattern analysis of StFH genes in drought stress treatments. According to your comment, we have redesign our RNA-seq reads pipeline based on STAR splice-aware aligner and used this STAR aligner for mapping the raw reads of of RNA-seq reads to the S. tuberosum genome. Surprisingly, we have got excellent results were using the STAR splice-aware aligner for mapping the raw reads of of RNA-seq reads to the S. tuberosum genome. We are very pleased to your kind and valuable suggestions and comments regarding splice-aware mappers (e.g., TopHat2/HISAT2, STAR). We also acknowledge your great suggestions which was unknown to us. Next time, we will use these splice-aware mappers. According to the new analysis results, we have revised the methods and materials as well as results and discussion section of expression pattern analysis of StFH genes. Please see the revised Fig. 16 and lines 208-214.

Comment #6: In Fig.1, the bootstrapping value should be added in the branch of the maximum likelihood (ML) tree. Compared to MEGA, IQ-TREE or RAxML might be a more appropriate tool for reconstructing ML tree, especially in the choice of the best-fitting substitution model.

Responses to Comment #6: Thank you for your valuable comment on Fig. 1. According to your comment, we have used the IQ tree version 1.6.12 with default parameters (http://www.iqtree.org/) to construct the phylogenetic tree. Also, we have added the bootstrapping value in the branch of the maximum likelihood (ML) tree according to your kind comment. According to the new analysis results, we have revised the methods and materials as well as results and discussion section of phylogenetic tree analysis. Please see the revised Fig. 1 and lines120-124,

Comment #7: In Fig.5, StFH20 and StFH22 were identified as a duplicated gene pair. It might be problem, owing to a great difference in their CDS length (Fig.3, > 3 fold). DupGen_finder or MCScanX should be a toolkit for detection of duplicated gene pairs.

Responses to Comment #7: Thank you very much for your kind and valuable comment on Fig. 5. According to your comment, we have used MCScanX toolkit for detection of duplicated gene pairs. Surprisingly, we have got good results by using MCScanX toolkit instead of our used previous analysis technique. Again, thanks a lot for providing us an incredible valuable suggestion and comment for the analysis of duplicated gene pairs. According to the new analysis results, we have revised the results and discussion section of Synonymous (Ks) and non-synonymous (Ka) substitution ratios calculation of StFH genes. Please see the revised Fig. 5 and line 145-146.

Comment #8: The gene name should be italicized, e.g., StFH (Line36, L40, L848, L851, L852, L854).

Responses to Comment #8: Thank you so much for your valuable comment. According to your comment, we have italicized all gene names in the whole manuscript.

Reviewer #2:

The paper titled: " Genome-wide identification and characterization of FORMIN gene family in potato (Solanum tuberosum L.) and their expression profiles in response to drought stress condition " reported the identification, characterization and of FORMIN gene family in Solanum tuberosum and the expression patterns of FORMIN members using RNA-seq data, offering a foundation for further research aiming at potato enhancement. Overall, the paper is clearly presented. The authors are invited to revise the English and the sentences structure in the manuscript before publication. Also, genes must be written in Italic in the whole text.

Comment #1: The authors are invited to revise the English and the sentences structure in the manuscript before publication. Also, genes must be written in Italic in the whole text.

Responses to Comment #1: Thank you for your valuable suggestion to improve our manuscript for language usage, spelling, and grammar by getting English editing and proofreading services. We are pleased to inform you that, to improve our manuscript at a standard level and to avoid grammatical errors, we are thoroughly edited and proofreading of our full manuscript by an academic English professional named Mr. Tanzir Ahmed, Assistant Professor, Department of English, Faculty of Arts and Social Science, Jashore University of Science and Technology, Jashore 7408, Bangladesh (https://just.edu.bd/t/teacher-1549444621686), email: tanzir@just.edu.bd which are highlighted as a track change options (Please see the track changes area around the full manuscript). We also acknowledged his name in the acknowledgement section of our manuscript (Please see the lines 833-836). Hopefully, the language of the revised manuscript is now more standard than the previous one. Moreover, according to your valuable comment, we have carefully checked and italicized all gene names in the whole manuscript.

---

## [Decision Letter · Decision Letter 1]

24 Jun 2024

PONE-D-24-14310R1Genome-wide identification and characterization of FORMIN gene family in potato (Solanum tuberosum L.) and their expression profiles in response to drought stress conditionPLOS ONE

Dear Dr. Sarkar,

Thank you for submitting your manuscript to PLOS ONE. After careful consideration, we feel that it has merit but does not fully meet PLOS ONE’s publication criteria as it currently stands. Therefore, we invite you to submit a revised version of the manuscript that addresses the points raised during the review process.

We look forward to receiving your revised manuscript.

Kind regards,

Muhammad Anwar, PHD

Academic Editor

PLOS ONE

Reviewers' comments:

Reviewer's Responses to Questions

**Comments to the Author**

1. If the authors have adequately addressed your comments raised in a previous round of review and you feel that this manuscript is now acceptable for publication, you may indicate that here to bypass the “Comments to the Author” section, enter your conflict of interest statement in the “Confidential to Editor” section, and submit your "Accept" recommendation.

Reviewer #1: (No Response)

Reviewer #2: All comments have been addressed

2. Is the manuscript technically sound, and do the data support the conclusions?

Reviewer #1: Yes

Reviewer #2: Yes

3. Has the statistical analysis been performed appropriately and rigorously? 

Reviewer #1: Yes

Reviewer #2: Yes

4. Have the authors made all data underlying the findings in their manuscript fully available?

Reviewer #1: Yes

Reviewer #2: Yes

5. Is the manuscript presented in an intelligible fashion and written in standard English?

Reviewer #1: Yes

Reviewer #2: No

6. Review Comments to the Author

Reviewer #1: Frist of all, thanks to the authors for their efforts in revising this manuscript. However, it is very unfortunate to know that these experiments (Comment #2, #3) could not be carried out due to lack of funds.

Please add literature (Line150).

Reviewer #2: The quality of the manuscript is clearly more better with this revision but a number of english errors are still detectible in the text. the text must be throughly revised to be free of english mistakes

7. PLOS authors have the option to publish the peer review history of their article (what does this mean?). If published, this will include your full peer review and any attached files.

Reviewer #1: No

Reviewer #2: No

---

## [Author Response · Author response to Decision Letter 1]

6 Jul 2024

Date: July 07, 2024

To

Professor Dr. Muhammad Anwar

Academic Editor

PLOS ONE

Ref: Manuscript ID PONE-D-24-14310R1

Subject: Submission of reviewer comments on the submitted manuscript in “PLOS ONE"

Thank you very much for your email dated June 24, 2024 and valuable time to go through the manuscript and point out some important issues. We would also like to take this opportunity to thank the editor and reviewer for giving their time and reviewing the manuscript and we appreciate the constructive comments and suggestions of the editor and the reviewer. 

We believe that such types of comments help us to make our manuscript scientifically sound. As per the editor and reviewer’s comments and suggestions, we have revised the manuscript entitled "Genome-wide identification and characterization of FORMIN gene family in potato (Solanum tuberosum L.) and their expression profiles in response to drought stress condition" and Manuscript ID PONE-D-24-14310R1.

We have responded to the concerns of the editor and the referee(s). Comments and responses/actions taken are shown in blue color font. The modified portion in the manuscript were highlighted with “Track Changes Option” in the revised manuscript.

Academic Editor suggestions:

• Suggestions#1: A rebuttal letter that responds to each point raised by the academic editor and reviewer(s). You should upload this letter as a separate file labeled 'Response to Reviewers'.

• Suggestions#2: A marked-up copy of your manuscript that highlights changes made to the original version. You should upload this as a separate file labeled 'Revised Manuscript with Track Changes'.

• Suggestions#3: An unmarked version of your revised paper without tracked changes. You should upload this as a separate file labeled 'Manuscript'.

Response to Academic Editor suggestions:

• Response to suggestions#1: We have made a ''Response to Reviewers' file to the academic editor and reviewer’s comments which are mentioned in the 'Reviewers report' section and revised the manuscript according to the point raised by the reviewer’s comments to the academic editor and reviewers. We have uploaded this file as a separate file labeled ''Response to Reviewers''.

• Response to suggestions#2: We have prepared a marked-up copy of our manuscript that is highlighted as a 'Track Changes' change made to the original version. We have uploaded this file as a separate file labeled 'Revised Manuscript with Track Changes'.

• Response to suggestion#3: We also keep an unmarked version of our revised manuscript without tracked changes. We have uploaded this file as a separate file labeled 'Manuscript'.

I hope we have tried to resolved the issues raised by the reviewer#1 and reviewer#2. However, if you need further clarification, please contact us without any hesitation.

Thank you very much in anticipation.

Kind regards,

Corresponding Author,

Dr. Md Abdur Rauf Sarkar (M.S. and Ph. D, Saga University, Japan)

Associate Professor

Laboratory of Functional Genomics and Proteomics

Department of Genetic Engineering and Biotechnology

Faculty of Biological Science and Technology

Jashore University of Science and Technology

Jashore-7408, Bangladesh 

Email: rauf.gebt@yahoo.com, mar.sarkar@just.edu.bd

Cell Phone: +880-1721-704800

Response to Reviewers

Reviewers' comments:

Responses to Reviewer's Questions

Review Comments to the Author

Reviewer #1:

Comment #1: Frist of all, thanks to the authors for their efforts in revising this manuscript. However, it is very unfortunate to know that these experiments (Comment #2, #3) could not be carried out due to lack of funds.

Please add literature (Line150).

Responses to Comment #1: Thank you for your valuable suggestion and consideration for your previous comments (Comment #2, #3) regarding our limitation in wet lab validation due to our lack of research funds. In this revision, according to your suggestion, we have added the literature in Line 150. Please see the Line150.

Reviewer #2:

Comment #1: The quality of the manuscript is clearly more better with this revision but a number of english errors are still detectible in the text. the text must be throughly revised to be free of english mistakes.

Responses to Comment #1: Thank you for your valuable suggestion to improve our manuscript avoiding language grammar errors, and common mistakes. According to your kind suggestions, in this revision, we have vigorously revised our manuscript to avoid grammatical errors, and common mistakes as well as tried to improve our manuscript at a standard level. In this revision, we also take the English language editing and proofreading opportunity from our academic English professional named Mr. Tanzir Ahmed, Assistant Professor, Department of English, Faculty of Arts and Social Science, Jashore University of Science and Technology, Jashore 7408, Bangladesh (https://just.edu.bd/t/teacher-1549444621686), email: tanzir@just.edu.bd which are highlighted as a track change options (Please see the track changes area around the full manuscript). We also acknowledged his name in the acknowledgment section of our manuscript (Please see lines 766-769). Hopefully, the language of the revised manuscript is now more standard than the previous one and you will consider it.

---

## [Decision Letter · Decision Letter 2]

11 Jul 2024

PONE-D-24-14310R2Genome-wide identification and characterization of FORMIN gene family in potato (Solanum tuberosum L.) and their expression profiles in response to drought stress conditionPLOS ONE

Dear Dr. Sarkar,

Thank you for submitting your manuscript to PLOS ONE. After careful consideration, we feel that it has merit but does not fully meet PLOS ONE’s publication criteria as it currently stands. Therefore, we invite you to submit a revised version of the manuscript that addresses the points raised during the review process.

We look forward to receiving your revised manuscript.

Kind regards,

Muhammad Anwar, PHD

Academic Editor

PLOS ONE

Reviewers' comments:

Reviewer's Responses to Questions

**Comments to the Author**

1. If the authors have adequately addressed your comments raised in a previous round of review and you feel that this manuscript is now acceptable for publication, you may indicate that here to bypass the “Comments to the Author” section, enter your conflict of interest statement in the “Confidential to Editor” section, and submit your "Accept" recommendation.

Reviewer #1: All comments have been addressed

Reviewer #2: All comments have been addressed

2. Is the manuscript technically sound, and do the data support the conclusions?

Reviewer #1: Partly

Reviewer #2: Yes

3. Has the statistical analysis been performed appropriately and rigorously? 

Reviewer #1: Yes

Reviewer #2: Yes

4. Have the authors made all data underlying the findings in their manuscript fully available?

Reviewer #1: Yes

Reviewer #2: Yes

5. Is the manuscript presented in an intelligible fashion and written in standard English?

Reviewer #1: Yes

Reviewer #2: Yes

6. Review Comments to the Author

Reviewer #1: [1] The expression pattern of FORMIN genes in Solanum tuberosum under drought stress is a main content of this manuscript entitled ‘Genome-wide identification and characterization of FORMIN gene family in potato (Solanum tuberosum L.) and their expression profiles in response to drought stress condition’. In addition, RT-qPCR results have been provided in a few recently published papers, such as Zhang et al. (PMID: 38256829) and Duan et al. (PMID: 34303338). Thus, it is necessary to provide the tissue/organ-specific and drought-responsive expression pattern of StFH genes using RT-qPCR.

[2] Solanum tuberosum is composed of two subspecies, such as tuberosum and andigena. There are a few potato genomes available at SpudDB (http://spuddb.uga.edu/), such as S. tuberosum cv. Otava (autotetraploid), S. tuberosum Group Phureja DM 1-3 516 R44 (doubled monoploid), and S. tuberosum group Tuberosum RH89-039-16 (heterozygous diploid). Thus, it is very fascinating how StFH genes are distributed on multiple different subgenomes of S. tuberosum.

[3] It is prerequisite to determine the syntenic relationship of FH genes in Solanum tuberosum with the other three species, including Arabidopsis thaliana, Oryza sativa and Zea may, such as the Figure 5 in the paper of Zhang et al. (PMID: 38256829). However, the Fig7 in this manuscript does not seem to reveal the collinear relationship of FH genes between Solanum tuberosum with the other three species. Thus, the interspecific synteny relationship of FH genes should be redrawn via TBtools (https://github.com/CJ-Chen/TBtools-II) or JCVI (https://github.com/tanghaibao/jcvi).

[4] There are still a lot of writing errors.

a) Writing errors in the Abstract section: ‘processes’ was falsely written as ‘processess’(Line26); ‘a genome-wide analysis of’ should be rewritten as ‘no genome-wide analysis of’ (Line27); ‘CAREs’ should be rewritten as ‘cis-acting regulatory elements (CAREs)’ (Line36); ‘uncovered’ was mistakenly written as ‘uncoverd’ (Line41).

b) In the Line173, ‘PlantRegMap’ was irrelevantly written as ‘PlantTFDB’.

c) In the Line179, a full stop was lost in the sentence ‘numerous important transcription factors (TFs). The collected data was subsequently visualized’.

d) In the Line490, ‘MicroRNAs’ was wrongly written as ‘MmicorRNAs’.

e) The genus name should be italicized, such as Arabidopsis (Line506, Line524, Line672).

f) In the Line718, ‘across’ should be revised as ‘in’.

Reviewer #2: The authors of the paper titled "Genome-wide identification and characterization of FORMIN gene family in potato

(Solanum tuberosum L.) and their expression profiles in response to drought stress

condition" have correctly adressed the required revisions

7. PLOS authors have the option to publish the peer review history of their article (what does this mean?). If published, this will include your full peer review and any attached files.

Reviewer #1: No

Reviewer #2: **Yes: **Hatem Boubakri

---

## [Author Response · Author response to Decision Letter 2]

4 Aug 2024

Response to Reviewers

Reviewers' comments:

Responses to Reviewer's Questions

Reviewer#1:

Comment#1: The expression pattern of FORMIN genes in Solanum tuberosum under drought stress is a main content of this manuscript entitled ‘Genome-wide identification and characterization of FORMIN gene family in potato (Solanum tuberosum L.) and their expression profiles in response to drought stress condition’. In addition, RT-qPCR results have been provided in a few recently published papers, such as Zhang et al. (PMID: 38256829) and Duan et al. (PMID: 34303338). Thus, it is necessary to provide the tissue/organ-specific and drought-responsive expression pattern of StFH genes using RT-qPCR.

Responses to Comment#1: Thank you for your valuable comment. According to your valuable comment, we have extracted two RNA-seq data for tissue/organ-specific and drought-responsive expression patterns of StFH genes. We have revised our manuscript based on those two data sets. Please see the details for tissue/organ-specific and drought-responsive expression patterns of StFH genes in the Materials and Methods section (Lines 200-220), Results section (Lines 500-535), Discussion section (Lines 642-666), Abstract section (Lines 43-47), Conclusion section (Lines 678-682). Thank you so much for providing the recent research articles related to expression analysis of this gene family members in other plant species.

In Review-1, Comment#2 of your Response to Comment#2, as we previously mentioned to you, we always appreciate your insights and fully agree with the importance of validating the candidate StFH genes using RT-qPCR through wet lab experiments. We would like to inform you that, currently we do not have a enough research grant to support the experimental validation of the data in our laboratory or to seek assistance from other well-equipped facilities. We sincerely apologize for the inconvenience caused regarding the RT-qPCR experiment and hope for your understanding and consideration regarding this issue.

Comment#2: Solanum tuberosum is composed of two subspecies, such as tuberosum and andigena. There are a few potato genomes available at SpudDB (http://spuddb.uga.edu/), such as S. tuberosum cv. Otava (autotetraploid), S. tuberosum Group Phureja DM 1-3 516 R44 (doubled monoploid), and S. tuberosum group Tuberosum RH89-039-16 (heterozygous diploid). Thus, it is very fascinating how StFH genes are distributed on multiple different subgenomes of S. tuberosum.

Responses to Comment#2: Thank you for your valuable comment. According to your nice and valuable comment, we have retrieved the genome information of three cultivars of Solanum tuberosum, S. tuberosum cv. Otava (autotetraploid), S. tuberosum Group Phureja DM 1-3 516 R44 (doubled monoploid), and S. tuberosum group Tuberosum RH89-039-16 (heterozygous diploid) from SpudDB (http://spuddb.uga.edu/) database and constructed a chromosomal distribution of StFH genes within the above three subgenomes of S. tuberosum. Hopefully, this StFH gene distribution information will enhance our manuscript quality. We will be grateful to you for your valuable comment on this. Please see the Fig 9 and Lines 157-163, 353-365, and 577-580. Again, thank you for your kind suggestion and provided us the important link.

Comment#3: It is prerequisite to determine the syntenic relationship of FH genes in Solanum tuberosum with the other three species, including Arabidopsis thaliana, Oryza sativa and Zea may, such as the Figure 5 in the paper of Zhang et al. (PMID: 38256829). However, the Fig7 in this manuscript does not seem to reveal the collinear relationship of FH genes between Solanum tuberosum with the other three species. Thus, the interspecific synteny relationship of FH genes should be redrawn via TBtools (https://github.com/CJ-Chen/TBtools-II) or JCVI (https://github.com/tanghaibao/jcvi).

Responses to Comment#3: Thank you for your valuable and kind comment. According to your valuable comment, we have constructed an interspecific synteny relationship of FH genes within three species such as Arabidopsis thaliana, Oryza sativa, and Zea mays including Solanum tuberosum using MCScanX of TB tools v.2.010 (https://github.com/CJ-Chen/TBtools-II). We hope this information will also increase our manuscript quality. Thank you so much for your nice comment. Please see the Fig 7 and Lines 147-151, 321-334, and 587-589.

Comment#4: There are still a lot of writing errors. a) Writing errors in the Abstract section: ‘processes’ was falsely written as ‘processess’(Line26 ==26); ‘a genome-wide analysis of’ should be rewritten as ‘no genome-wide analysis of’ (Line27 ==L26); ‘CAREs’ should be rewritten as ‘cis-acting regulatory elements (CAREs)’ (Line36 ==L36); ‘uncovered’ was mistakenly written as ‘uncoverd’ (Line41 ==Excluded).

Responses to Comment#4(a): Thank you for your valuable comments and suggestions. We have corrected the mentioned lines according to your comments. The word ‘uncovered’ is replaced with ‘identified’. Please see the lines 26, 26, 36, and 42.

Comment#4(b): In the Line173, ‘PlantRegMap’ was irrelevantly written as ‘PlantTFDB’

Responses to Comment#4(b): Thank you for your valuable comment. We have corrected the term as ‘PlantRegMap’. Please see the lines 174-175.

Comment#4(c): In the Line179, a full stop was lost in the sentence ‘numerous important transcription factors (TFs). The collected data was subsequently visualized’.

Responses to Comment#4(c): Thank you for your valuable comment. We have corrected it. Please see the line 181.

Comment#4(d): In the Line490, ‘MicroRNAs’ was wrongly written as ‘MmicorRNAs’.

Responses to Comment#4(d): Thank you for your valuable comment. We have corrected the term. Please see the line 468.

Comment#4(e): The genus name should be italicized, such as Arabidopsis (Line506, Line524, Line672).

Responses to Comment#4(e): Thank you for your valuable comment. We have italicized the genus name, Arabidopsis. Please see lines 480, 494, and 620.

Comment#4(f): In the Line718, ‘across’ should be revised as ‘in’.

Responses to Comment#4(f): Thank you for your valuable comment. We have replaced ‘across’ with ‘in’. Please see the line 669.

Reviewer #2:

Comment#1: The authors of the paper titled "Genome-wide identification and characterization of FORMIN gene family in potato (Solanum tuberosum L.) and their expression profiles in response to drought stress condition" have correctly adressed the required revisions

Responses to Comment#1: Thank you so much for giving your valuable time, kind comments, and suggestions to improve the quality of our manuscript. We are also very pleased to address your all previous comments and suggestions.

---

## [Decision Letter · Decision Letter 3]

12 Aug 2024

Genome-wide identification and characterization of FORMIN gene family in potato (Solanum tuberosum L.) and their expression profiles in response to drought stress condition

PONE-D-24-14310R3

Dear Dr. Abdur Rauf Sarkar,

We’re pleased to inform you that your manuscript has been judged scientifically suitable for publication and will be formally accepted for publication once it meets all outstanding technical requirements.

Kind regards,

Muhammad Anwar, PHD

Academic Editor

PLOS ONE

Additional Editor Comments (optional):

Reviewers' comments:

Reviewer's Responses to Questions

**Comments to the Author**

1. If the authors have adequately addressed your comments raised in a previous round of review and you feel that this manuscript is now acceptable for publication, you may indicate that here to bypass the “Comments to the Author” section, enter your conflict of interest statement in the “Confidential to Editor” section, and submit your "Accept" recommendation.

Reviewer #1: All comments have been addressed

Reviewer #3: All comments have been addressed

2. Is the manuscript technically sound, and do the data support the conclusions?

Reviewer #1: Yes

Reviewer #3: Yes

3. Has the statistical analysis been performed appropriately and rigorously? 

Reviewer #1: Yes

Reviewer #3: Yes

4. Have the authors made all data underlying the findings in their manuscript fully available?

Reviewer #1: Yes

Reviewer #3: Yes

5. Is the manuscript presented in an intelligible fashion and written in standard English?

Reviewer #1: Yes

Reviewer #3: Yes

6. Review Comments to the Author

Reviewer #1: I have no comment on this manuscript. Thanks to the authors for their efforts in revising the manuscript.

Reviewer #3: Author addressed all the suggested modifications and incorporated within the manuscript. accept for publishing.

7. PLOS authors have the option to publish the peer review history of their article (what does this mean?). If published, this will include your full peer review and any attached files.

Reviewer #1: No

Reviewer #3: No

---

## [Editor Report · Acceptance letter]

14 Aug 2024

PONE-D-24-14310R3 

PLOS ONE

Dear Dr. Sarkar, 

I'm pleased to inform you that your manuscript has been deemed suitable for publication in PLOS ONE. Congratulations! Your manuscript is now being handed over to our production team.

Kind regards, 

on behalf of

Dr. Muhammad Anwar 

Academic Editor

PLOS ONE